# PoseX: AI Defeats Physics-based Methods on Protein Ligand Cross-Docking

**Yize Jiang**[1,*]**, Xinze Li**[1,*]**, Yuanyuan Zhang**[2,*]**, Jin Han**[3,*]**, Youjun Xu**[4,*]**, Ayush Pandit**[5]**,
Zaixi Zhang**[6]**, Mengdi Wang**[6]**, Mengyang Wang**[7]**, Chong Liu**[8]**, Guang Yang**[9]**, Yejin Choi**[5]**,
Yingzhou Lu**[5]**, Wu-Jun Li**[3,†]**, Tianfan Fu**[3,†]**, Fang Wu**[5,†]**, Junhong Liu**[1,†,*]

[1]Microcyto, [2]Purdue University, [3]State Key Laboratory for Novel Software Technology at
Nanjing University, [4]Anew Therapeutics, [5]Stanford University, [6]Princeton University,
[7]Peking University, [8]Central South University, [9]Imperial College London
[*]Equal Contributions, [†]Correspondence

## Abstract

Recently, significant progress has been made in protein-ligand docking, particularly in deep learning methods, and benchmarks have been proposed, such as PoseBench and PLINDER. However, these studies typically focus on the self-docking scenario, which is less practical in real-world applications. Moreover, some studies employ complex frameworks that require extensive training, posing challenges for convenient and efficient assessment of docking methods. To address these gaps, we introduce PoseX, an open-source benchmark for evaluating both self-docking and cross-docking, enabling a practical and comprehensive assessment of algorithmic advances. Specifically, we curated a novel dataset comprising 718 entries for self-docking and 1,312 entries for cross-docking; secondly, we incorporated 23 docking methods in three methodological categories, including *physics-based methods* (e.g., Schrödinger Glide), *AI docking methods* (e.g., DiffDock) and *AI co-folding methods* (e.g., AlphaFold3); thirdly, we developed a relaxation method for post-processing to minimize conformational energy and refine binding poses; fourthly, we established a public leaderboard to rank submitted models in real-time. We derived some key insights and conclusions through extensive experiments: (1) AI-based approaches consistently outperform *physics-based methods* in overall docking success rate. (2) Most intra- and intermolecular clashes of AI-based approaches can be greatly alleviated with relaxation, which means combining AI modeling with physics-based post-processing could achieve excellent performance. (3) *AI co-folding methods* exhibit ligand chirality issues, except for Boltz-1x, which introduced physics-inspired potentials to fix hallucinations, suggesting that stereochemical modeling greatly improves the structural plausibility of the predicted protein-ligand complexes. (4) Specifying binding pockets significantly promotes docking performance, indicating that pocket information can be leveraged adequately, particularly for *AI co-folding methods*, in future modeling efforts.

🧩 http://dock-lab.tech/

⭘ https://github.com/CataAI/PoseX

🤗 https://huggingface.co/datasets/CataAI/PoseX

## 1 Introduction

Protein-ligand docking is crucial to drug discovery as it predicts how a ligand interacts with a protein, helping to identify potential drug candidates and accelerate the development of new therapeutics Huang et al. (2022); Du et al. (2022); Fu et al. (2022); Wu et al. (2023; 2024). By understanding these interactions, researchers can optimize ligands for better binding affinity, specificity, and efficacy, ultimately accelerating the development of new therapeutics. Learning from known crystal protein-ligand complexes through machine learning, especially deep learning (DL) techniques, AI-based approaches have revolutionized protein-ligand docking and substantial progress has been made recently (Pei et al., 2023; Lu et al., 2024; Lai et al., 2024; Cao et al., 2024). In response

Table 1: Comparison of existing docking benchmark studies.

| Benchmarks | PoseBuster | PoseBench | PLINDER | PoseX (Ours) |
|---|---|---|---|---|
| Code of dataset pipeline | ✗ | ✗ | ✓ | ✓ |
| Relaxation | ✗ | coarse | ✗ | well-designed |
| Self-docking evaluation | ✓ | ✓ | ✓ | ✓ |
| Cross-docking evaluation | ✗ | ✗ | ✗ | ✓ |
| # Open-source docking software | 2 | 1 | 0 | 2 |
| # Commercial docking software | 0 | 0 | 0 | 3 |
| # *Physics-based methods* | 2 | 1 | 0 | 5 |
| # *AI docking methods* | 5 | 2 | 1 | 11 |
| # *AI co-folding methods* | 0 | 4 | 0 | 7 |
| # Total methods | 7 | 7 | 1 | 23 |
| Real-time leaderboard | ✗ | ✗ | ✗ | ✓ |

to the proliferation of new approaches, recent work has introduced several benchmarks, such as PoseBench (Morehead et al., 2025) and PLINDER (Durairaj et al., 2024), along with corresponding datasets and metrics for evaluating protein-ligand interactions. Despite the rapid progress, existing studies still encounter several challenges, summarized as follows.

1. **Self-docking is an impractical setup.** Most existing benchmarks, such as PoseBuster (Buttenschoen et al., 2024) and PoseBench (Morehead et al., 2025), focus on the self-docking scenario, which is less practical in real-world applications. For instance, pharmaceutical chemists typically design new drug molecules and dock them into targets, whose conformations are extracted from existing crystal structures co-crystallized with other published compounds.

2. **Heavy framework and low accessibility.** Some benchmarks (e.g., PLINDER (Durairaj et al., 2024)) suffer from heavy evaluation frameworks that involve data splitting and training, which are hard to use. While studies such as PoseBuster thafocus solely on evaluation rather than training are valuablece, they are lightweight and user-friendly.

3. **Limited model selection for benchmarking.** Existing studies often restrict their comparative scope to a narrow set of models. For instance, PoseBuster evaluated only 5 AI-based approaches and 2 *physics-based methods*, while PLINDERwhereas PLINDER benchmarked only against DiffDock (Corso et al., 2022), thereby her notable algorithms.

Therefore, we propose several solutions to address these issues:

1. **Cross-docking is more realistic.** To better evaluate the capacity of various docking methods in a more practical scenario, incorporate cross-docking, which involves docking various small molecules extracted from distinct complexes of the same protein with all the conformations except the native co-crystalized one.

2. **Construction of new dataset.** We curated a new dataset named PoseX that collects newly found crystal structures of protein-ligand complexes in RCSB PDB, which contains 718 entries for self-docking and 1,312 entries for cross-docking.

3. **Involving 20+ models.** We evaluated 23 docking methods encompassing nearly all relevant models published in peer-reviewed journals and conferences alongside established commercial docking software across three different categories, including 5 *physics-based methods* such as Schrödinger Glide (Friesner et al., 2004), 11 *AI docking methods* such as DiffDock, and 7 *AI co-folding methods* such as AlphaFold3 (Abramson et al., 2024).

In addition, we developed a novel *relaxation* module (also known as *energy minimization*) that serves as a post-processing step to refine AI-generated binding poses and improve structural plausibility. We also established a public online leaderboard that enables researchers to benchmark their models against a standardized dataset, fostering transparency and facilitating straightforward, fair comparisons for the broader community. The key differences between the existing docking benchmarks and ours are summarized in Table 1.

Table 2: Comparison of various docking methods.

| Method | Pub. Year | License | Pocket Required | Pocket Changed | Avg. Runtime Per Sample [1] |
|---|---|---|---|---|---|
| **Physics-based methods** | | | | | |
| Discovery Studio | late 1990s | Commercial | ✓ | ✗ | 14.4 min |
| Schrödinger Glide | 2004 | Commercial | ✓ | ✗ | 7.2 min |
| MOE | 2008 | Commercial | ✓ | ✗ | 50 sec |
| AutoDock Vina | 2010, 2021 | Apache-2.0 | ✓ | ✗ | 18 sec |
| GNINA | 2021 | Apache-2.0 | ✓ | ✗ | 12 sec |
| **AI docking methods** | | | | | |
| DeepDock | 2021 | MIT | ✓ | ✗ | 2.7 min |
| EquiBind | 2022 | MIT | ✗ | ✗ | 1.4 sec |
| TankBind | 2022 | MIT | ✗ | ✗ | 7.8 sec |
| DiffDock | 2022 | MIT | ✗ | ✗ | 1.2 min |
| UMD V2 | 2024 | MIT | ✓ | ✗ | 24 sec |
| FABind | 2023 | MIT | ✗ | ✗ | 8.8 sec |
| DiffDock-L | 2024 | MIT | ✗ | ✗ | 1.5 min |
| DiffDock-Pocket | 2024 | MIT | ✓ | ✓ | 1.7min |
| DynamicBind | 2024 | MIT | ✗ | ✓ | 2.4 min |
| Interformer | 2024 | Apache-2.0 | ✓ | ✗ | 0.6 min |
| SurfDock | 2024 | MIT | ✓ | ✗ | 10.8 sec |
| **AI co-folding methods** | | | | | |
| NeuralPLexer | 2024 | BSD | ✗ | ✓ | 1.5 min |
| RFAA | 2023 | BSD | ✗ | ✓ | 9 min |
| AlphaFold3 | 2024 | CC-BY-NC-SA 4.0 | ✗ | ✓ | 16.5 min |
| Chai-1 | 2024 | Apache-2.0 | ✗ | ✓ | 3 min |
| Boltz-1 | 2024 | MIT | ✗ | ✓ | 3 min |
| Boltz-1x | 2025 | MIT | ✗ | ✓ | 3 min |
| Protenix | 2025 | Apache-2.0 | ✗ | ✓ | 3.6 min |

[1] The running environment and parameters of each method are shown in Appendix B.

## 2 METHODS

We categorize all the docking approaches into three distinct categories: (1) *physics-based methods* utilize physics-based scoring functions and sampling algorithms to estimate protein-ligand interactions, including Discovery Studio (Pawar & Rohane, 2021), Schrödinger Glide (Friesner et al., 2004), MOE (Vilar et al., 2008), AutoDock Vina (Trott & Olson, 2010; Eberhardt et al., 2021), and GNINA (McNutt et al., 2021); (2) *AI docking methods* produce ligand binding poses based on the three-dimensional structure of proteins, including DeepDock (Méndez-Lucio et al., 2021), EquiBind (Stärk et al., 2022), TankBind (Lu et al., 2022), DiffDock (Corso et al., 2022), Uni-Mol Docking V2 (UMD V2) (Alcaide et al., 2024), FABind (Pei et al., 2023), DiffDock-L (Corso et al., 2024), DiffDock-Pocket (Plainer et al., 2023), DynamicBind (Lu et al., 2024), Interformer (Lai et al., 2024), SurfDock (Cao et al., 2024); (3) *AI co-folding methods* predict both the ligand's binding conformation and the protein's conformational changes induced by ligand binding, which account for simultaneous structural adaptations of the protein and ligand, enabling more accurate modeling of their interactions; we involve 7 *AI co-folding methods*, including NeuralPLexer (Qiao et al.), RoseTTAFold-All-Atom (RFAA) (Krishna et al., 2024), AlphaFold3 (Abramson et al., 2024), Chai-1 (Discovery et al., 2024), Boltz-1 (Wohlwend et al., 2024), Boltz-1x (Wohlwend et al., 2024), Protenix (Team et al., 2025). For comparative analysis, we summarize the methods compared in Table 2, and the detailed settings of these methods are provided in Appendix B.

**Relaxation as Post-processing** Relaxation in molecular docking, also known as *energy minimization*, is a post-processing method used to refine and optimize docked protein-ligand complexes (Guedes et al., 2014; Amaro et al., 2008). It involves energy minimization and, when necessary, short molecular dynamics simulations to resolve steric clashes, refine interatomic interactions, and ensure that the system reaches a stable, low-energy conformation. This step enhances the physical realism and accuracy of the docking results, thereby making the predicted binding poses more reliable for subsequent analysis or experimental validation. In this paper, we introduce a well-designed relaxation

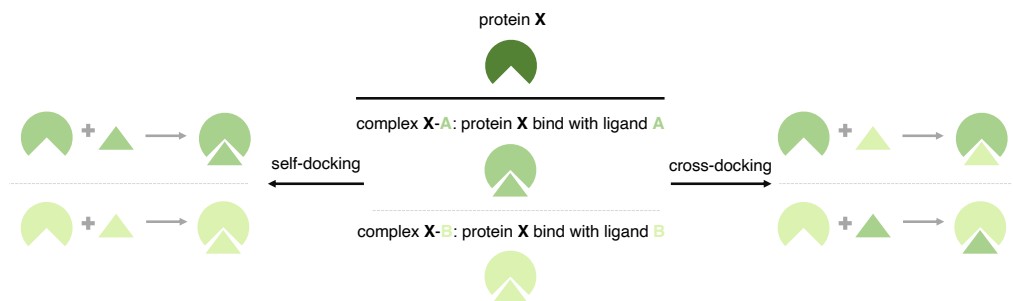

Figure 1: Illustration of two docking setups: (1) self-docking, vs. (2) cross-docking.

module, the feature of which is summarized as: (1) Implemented an automated relaxation process for complexes based on OpenMM (Eastman et al., 2017). (2) Established a comprehensive automatic data processing pipeline for proteins and small molecules, including fixing missing chains, capping the N- and C-termini, adding formal charges to proteins and small molecules, and applying restraints to backbone atoms (CA, C, N, O). (3) Supports small molecule force field parameters from GAFF and OpenFF (Consortium, 2024). (4) Supports partial charge calculation methods for small molecules, including Gasteiger and MMFF94. (5) Effectively alleviates unreasonable predicted conformations, improving the pass rate of PB-Valid. The technical details of the relaxation process are provided in Appendix C.

## 3 DATASET

### 3.1 SELF-DOCKING VERSUS CROSS-DOCKING

**Self-docking**. Self-docking involves docking a ligand back into its native co-crystallized conformation (Kawatkar et al., 2009). This is typically used to assess whether the docking software accurately reproduces the known binding pose, thereby validating the method. Most existing benchmarks only consider the self-docking setup.

**Cross-docking**. Cross-docking refers to dock molecules extracted from distinct complexes of the same protein with all conformations except the native co-crystalized one. This approach is considered more versatile, as it accounts for the possibility that the receptor protein may undergo conformational changes and may not be fully optimized for ligand docking. The difference between self-docking and cross-docking is illustrated in Figure 1.

### 3.2 ASTEX

The Astex Diverse set (Hartshorn et al., 2007), published in 2007, is a set of hand-picked, relevant, diverse, and high-quality protein–ligand complexes from the RCSB PDB. It comprises 85 unique and significant protein-ligand complexes. These complexes have been appropriately formatted for docking purposes and will be made freely accessible to the entire research community via the website (http://www.ccdc.cam.ac.uk). The Astex Diverse set only supports self-docking evaluation.

### 3.3 POSEX: OUR CURATED DATASET

In this paper, we curated a high-quality protein-ligand complex structure dataset, PoseX, designed to evaluate molecular docking methods. It comprises carefully selected crystal structures from the RCSB Protein Data Bank (RCSB PDB) (Rose et al., 2016), with two subsets for evaluating self-docking and cross-docking tasks. The dataset includes only complex structures published from 2022 to January 1st, 2025, ensuring no overlap with the training data of all AI-based approaches being evaluated (as shown in Table S3). The construction steps of the two subsets PoseX Self-Docking (PoseX-SD) and PoseX Cross-Docking (PoseX-CD) are shown in Table S1 and Table S2. Ultimately, there are 718 entries for PoseX-SD and 1,312 entries for PoseX-CD, comprising 109 protein targets (a total of 371 structures) and 362 small molecules. The distribution of the number of conformation structures per target is shown in Figure S1a, and the distribution of pocket similarity is shown in Figure S1b.

# 4 EXPERIMENTS

## 4.1 EVALUATION METRICS

Performance evaluation of protein-ligand docking involves metrics that assess both the quality of the predicted binding pose and the chemical validity as well as the structural plausibility, which are described in detail as follows.

**RMSD**. In accordance with most benchmarking studies, we evaluate the quality of binding poses using the Root Mean Square Deviation (RMSD), which measures the distance between the predicted and ground-truth complex structures. Lower RMSD scores indicate better binding poses.

**PB-Valid**. The physicochemical validity and structural plausibility of the generated binding poses are measured with the PoseBuster test suite (i.e., PB-Valid). This suite evaluates whether predicted ligand poses are consistent with known chemical and structural constraints. See Appendix D for more details.

**Success rate**. The docking success rate is defined as the percentage of the top-1 ranked predictions satisfying either of the following criteria: (1) RMSD < 2Å, or (2) RMSD < 2Å & PB-Valid. For PoseX-CD, we report the averaged success rate at the target level in view of the uneven distribution of docking sizes per target (as shown in Figure S1a). Higher success rates indicate better performance.

## 4.2 OVERALL PERFORMANCE ANALYSIS

Figure 2, Figure 3, and Table S4 present a comprehensive evaluation of various docking approaches on three benchmarks — PoseX-SD, PoseX-CD, and Astex — under RMSD < 2Å and PB-Valid criteria. From these results, we highlight several key observations and provide a more detailed analysis.

1. **AI-based approaches lead in success rate.** The latest AI-based approaches, both *AI docking methods* (e.g., SurfDock) and *AI co-folding methods* (e.g., AlphaFold3) have consistently outperformed *physics-based methods* in overall docking pose and validity.

2. **Relaxation significantly mitigates clashing.** The intra- and intermolecular clashes of AI-based approaches can be greatly alleviated with relaxation, which means that the force field-based energy minimization step is very crucial to achieve excellent performance in real-world applications, particularly for AI modeling.

3. **Chirality warrants further improvement.** Most of the *AI co-folding methods* exhibit ligand chirality issues, such as AlphaFold3 and Chai-1, except for Boltz-1x, which introduces an inference time steering technique employing physics-inspired potential to fix hallucinations and enhance structural plausibility.

4. **Pocket information is crucial to docking.** Explicit modeling of the binding pocket substantially improves docking performance, as seen by the consistent performance gains of DiffDock-Pocket over its counterpart DiffDock across both self-docking and cross-docking, indicating that pocket information can be leveraged adequately, especially for *AI co-folding methods*, in future modeling efforts.

**Astex Benchmark.** The Astex benchmark represents an idealized docking scenario with high-quality co-crystal structures. Because most AI-based approaches use the PDBBind v2020 training set, which includes 16,379 protein-ligand complexes, we analyzed and found that 43 of the 85 complexes in the Astex Diverse Set are included in this set. In this setting, *AI docking methods* outperform all other categories overall. UMD V2 and SurfDock achieve the highest docking success rates (94.1%) when integrated with our structural relaxation protocol, surpassing *physics-based methods*, such as Glide and Discovery Studio, by over 25%. DiffDock-Pocket, Interformer, and DiffDock-L also perform strongly, achieving success rates above 83.8%. While *AI co-folding methods* such as AlphaFold3, Protenix, and Chai-1 deliver competitive results (over 80% success), they are marginally outperformed by docking-specialized architectures. *Physics-based methods* like AutoDock Vina and MOE plateau around 56.4%–67.1%, even with induced-fit docking (e.g., Glide IFD). These results illustrate substantial performance gains from AI models tailored to pose prediction.

**PoseX-SD Benchmark.** For PoseX-SD evaluation, SurfDock (78.0%) achieves the overall state-of-the-art performance, and UMD V2 takes the second place. DiffDock-Pocket shows clear advantages

over its pocket-agnostic counterpart, with a success rate of 52.6%. Among *AI co-folding methods*, AlphaFold3 and Protenix perform well (60.5% and 56.3%, respectively), demonstrating their capacity to model close-range binding interactions. In contrast, earlier *AI docking methods* such as EquiBind and TankBind perform poorly (below 20%) and exhibit significant issues with structural plausibility. *Physics-based methods* such as Glide and Discovery Studio remain clustered in the 40–65% range. Most AI-based approaches benefit from the relaxation method we developed, and their intra- and intermolecular validity is significantly improved.

Taking into account that **pocket information** plays a key role in predicting binding structures, we further separate the methods into *Pocket-Given* (docking with specified pocket) and *Blind-Docking* (docking without specified pocket) tracks and analyze, respectively, as shown in Figure 3. In the *Pocket-Given* track, SurfDock leads with 78.0% success rate, outperforming UMD V2 (72.4%), Interformer (66.6%), and DiffDock-Pocket (52.6%). These methods leverage explicit pocket information to refine pose predictions, demonstrating superior accuracy in scenarios where binding sites are known a priori. *Physics-based methods* such as Glide (47.9%) and Discovery Studio (54.9% ) are classical computational chemistry docking tools which inherently require pocket specification, while they were all surpassed by the *AI docking methods*. In the *Blind-Docking* track, AlphaFold3 achieves the best success rate of 60.3%, followed by Protenix (56.3%) and Chai-1 (56.1%). These *AI co-folding methods* excel in modeling induced-fit adaptations without specification of pockets, solving a more challenging problem than *Pocket-Given* counterparts. Blind *AI docking methods* like DiffDock-L (47.1%) and DynamicBind (27.0%) perform well while still defeated by *AI co-folding methods*, which underscores the value of simultaneous protein-ligand interaction modeling in *Blind-Docking* scenarios.

**PoseX-CD Benchmark.** For PoseX-CD evaluation, SurfDock (77.0%) and UMD V2 (69.2%) are still the top performers in all three categories of docking methods, as well as AlphaFold3, which achieves competitive performance (68.6%) against UMD V2. We observed that *AI docking methods* have developed rapidly in recent years, of which the latest models (such as SurfDock, UMD V2, Interformer, and DiffDock-Pocket) demonstrably surpass the earlier models (such as EquiBind, TankBind, and DeepDock). For *AI co-folding methods*, AlphaFold3 defeats other models (such as Chai-1, Boltz-1, Boltz-1x and Proteinx) by a narrow margin (1.6% - 7.5%). Notably, *physics-based methods* struggle significantly in this scenario. For example, in the PoseX-SD task, only 3 *AI docking methods* outperform the leading *physics-based method*, GNINA, in terms of the percentage of RMSD < 2Å with relaxation. However, in the PoseX-CD task, 9 AI-based approaches (including 4 *AI docking methods* and 5 *AI co-folding methods*) surpass GNINA (54.1%). This underscores a significant advantage of AI-based approaches over *physics-based methods* in the cross-docking scenario. Figure S12 and Figure S13 depict an illustrative example of the superior performance of AI-based methods. Relaxation yields consistent improvements across most approaches, emphasizing its role in resolving steric or geometric inconsistencies.

Furthermore, we also analyzed the performance according to whether the **pocket information** is specified, as shown in Figure 3. In the *Pocket-Given* track, SurfDock is also the winner with the highest success rate of 77.0%, followed by UMD V2 (69.2%), Interformer (60.2%), and DiffDock-Pocket (58.5%). These methods benefit from pocket information to handle cross-conformational variability, outperforming *physics-based methods* like Discovery Studio (43.7%), Glide (38.4%), and MOE (33.3%). This track highlights the advantages of pocket-guided *AI docking methods* in practical drug design scenarios involving non-native protein conformations. In the *Blind-Docking* track, AlphaFold3 leads at 68.8%, with Chai-1 (67.0%) and Boltz-1x (64.4%) close behind. *AI co-folding methods* dominate here, as they inherently model protein flexibility without pocket information, addressing the harder blind-docking challenge. Blind *AI docking methods* such as DiffDock-L (54.7%) and DynamicBind (32.8%) showed excellent performance but were generally outpaced, emphasizing that it is necessary to incorporate co-folding-like adaptability for *Blind-Docking* methods in future.

### 4.3 Pocket Similarity based Generalizability Analysis

To further understand the generalization capacity of various docking approaches, we analyze the relationship between pocket similarity and ligand RMSD across different scenarios. In view of the cut-off time of the training data for each method (as shown in Table S3), the pocket similarity is calculated as the maximum TM-score compared to pockets extracted from crystal structures released before 2022 on RCSB PDB, where the pocket is defined as the residues within 10.0Å of the ligand.

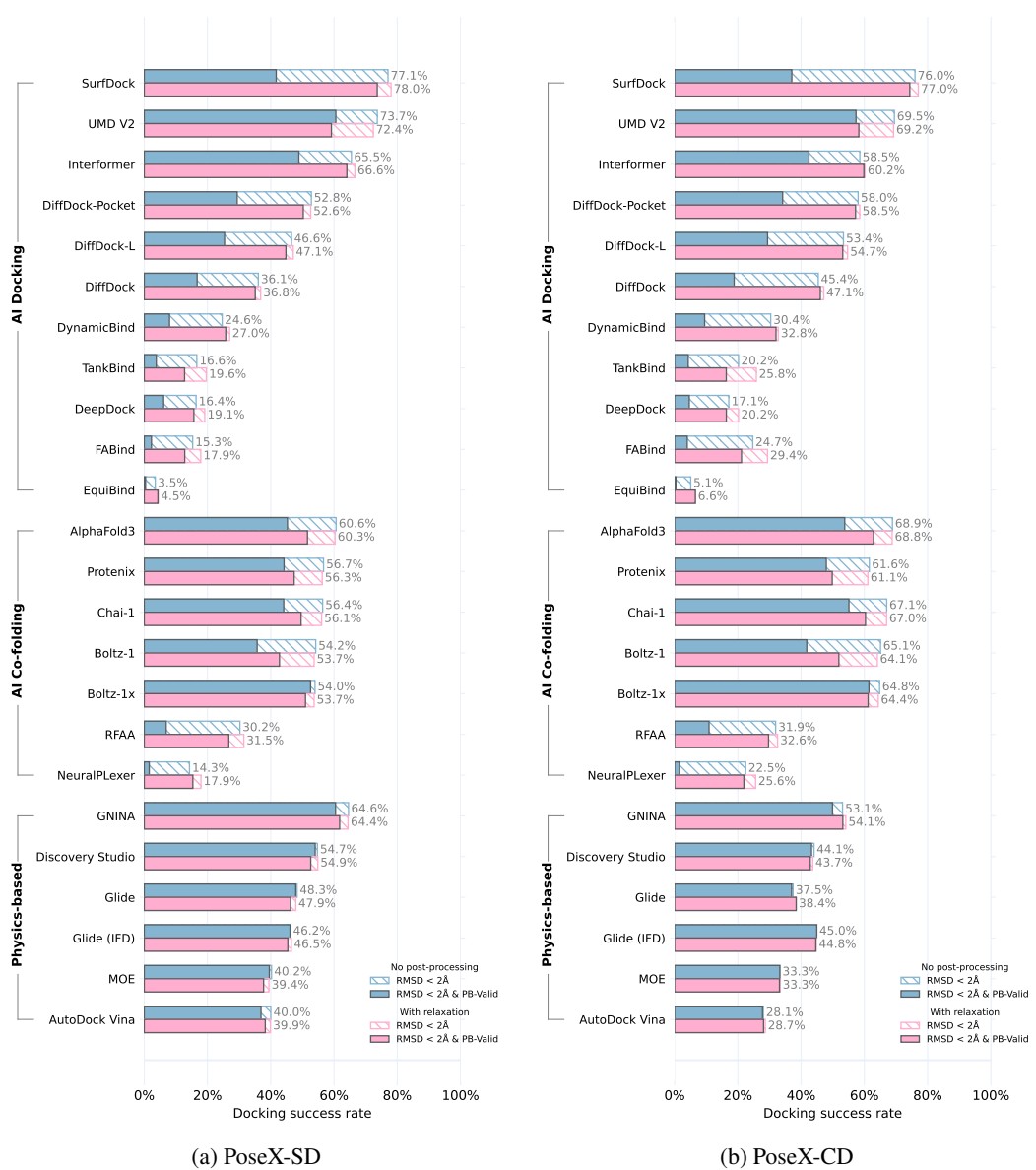

(a) PoseX-SD

(b) PoseX-CD

Figure 2: Performance on PoseX-SD and PoseX-CD. Mean values of three independent runs are reported here for each method, and detailed results with standard deviation are reported in Table S4. Striped bars represent the proportion of predictions with RMSD < 2Å, with numerical values indicated using bar labels. Solid bars indicate predictions that additionally satisfy PoseBuster validation criteria (PB-valid). Results with and without relaxation are distinguished by different colors.

Figure S2 and Figure S3 present per-sample scatter plots of pocket similarity versus docking RMSD for self-docking and cross-docking, respectively. Each plot reports Pearson's correlation coefficient to quantify the strength and direction of the relationship. Figure 4 complements these results by summarizing the average ligand RMSD separately for test cases with similar and dissimilar pockets. Figure S4 and Figure S5 illustrate the relationship between the ligand RMSD and the decreasing binding pocket similarity of AI-based approaches.

**Self-Docking Observations.** In the self-docking scenario, most AI-based approaches exhibit a moderate negative correlation between pocket similarity and ligand RMSD, indicating that the leakage of pocket information is associated with improved ligand pose accuracy. For example, Protenix and Chai-1 show stronger correlations ($r = -0.390$ and $r = -0.389$, respectively), while other models

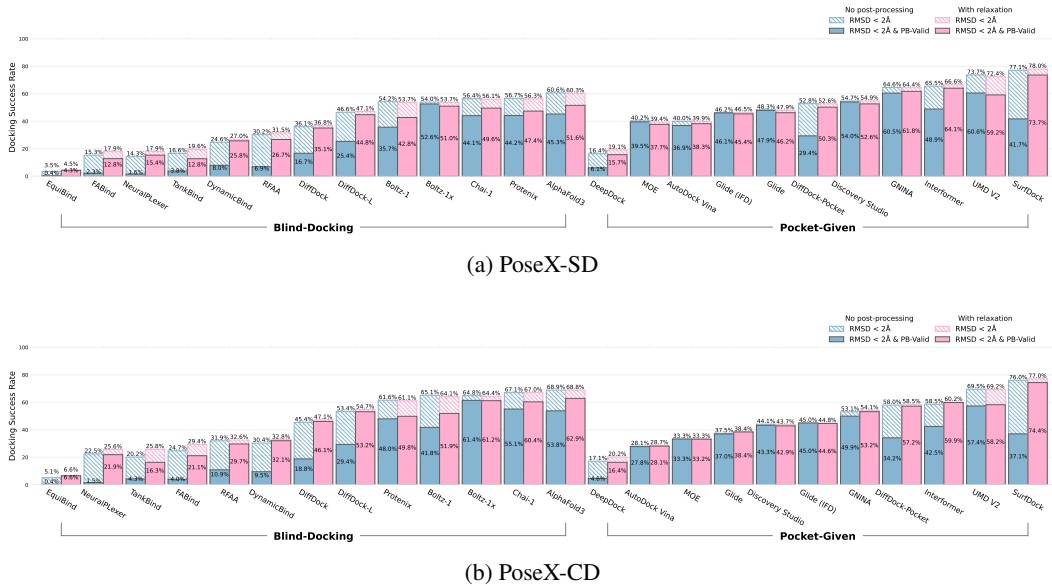

Figure 3: Performance of each model on the PoseX-SD and PoseX-CD datasets. Results are split according to whether the method requires a pocket (Pocket-Given vs. Blind-Docking).

such as AlphaFold3 ($r = -0.313$) and Boltz-1 ($r = -0.276$) exhibit similar trends. DiffDock and DiffDock-L display similar correlations ($r = -0.283$ and $r = -0.278$, respectively), suggesting that docking-specific models also benefit from the pocket leakage.

In contrast, *physics-based methods* show weaker or near-zero correlations. Glide ($r = 0.010$), AutoDock Vina ($r = -0.009$), and Discovery Studio ($r = -0.001$) exhibit negligible correlations, suggesting consistent docking performance across varying pocket similarities.

Notably, SurfDock ($r = -0.091$) and UMD V2 ($r = -0.134$), which achieve top overall performance, show only a weak correlation between pocket similarity and ligand RMSD. These findings suggest that their success likely stems from robust pose-prediction mechanisms that are less sensitive to pocket information leakage. These results highlight the importance of robust pose prediction in achieving high docking performance, even when pocket similarity is limited, in the self-docking scenario.

**Cross-Docking Observations.** The cross-docking setting reveals an overall stronger correlation between pocket similarity and ligand RMSD, particularly for *AI co-folding methods* and *AI docking methods*. Chai-1 ($r = -0.526$), Boltz-1 ($r = -0.521$), and Protenix ($r = -0.553$) exhibit strong negative correlations, suggesting that successful docking in cross-docking is highly contingent upon correctly modeling the target pocket's conformation. DiffDock and its variants continue to reflect this trend (e.g., DiffDock $r = -0.505$; DiffDock-L $r = -0.498$), further confirming the influence of pocket leakage under receptor shift scenarios.

Models such as DynamicBind ($r = -0.576$) and DiffDock-Pocket ($r = -0.425$) also show a strong correlation between pocket similarity and ligand RMSD, reinforcing that flexible or dynamic *AI docking methods* also have constrained generalization. In contrast, *physics-based methods* such as Glide ($r = 0.015$) and Discovery Studio ($r = 0.053$) again exhibit negligible correlation.

Even high-performing models like SurfDock ($r = -0.376$) and UMD V2 ($r = -0.280$) show stronger correlations in this setting than in self-docking, indicating that pocket modeling becomes more critical in the presence of conformational variance. This further highlights the need for improved pocket-conditioned pose generation in cross-docking scenarios.

**Performance Stratified by Pocket Similarity.** Figure 4 further stratifies the average ligand RMSD for each method, where the evaluation set is split into two groups, a similar group (96 protein targets, Pocket Similarity $\geq 0.70$) and a dissimilar group (13 protein targets, Pocket Similarity

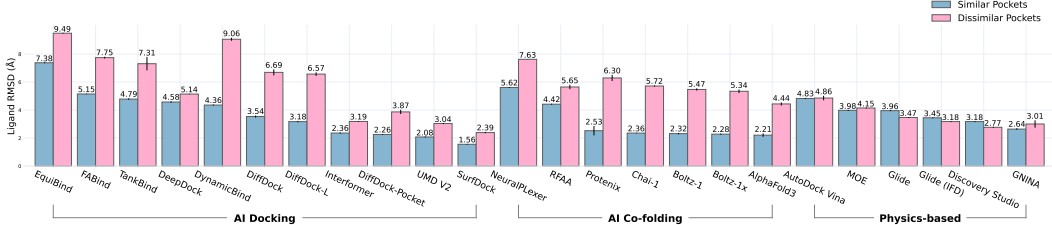

Figure 4: Cross-docking performance difference on "similar" and "dissimilar" binding pockets.

< 0.70). Across all the AI-based approaches, both *AI docking methods* and *AI co-folding methods*, docking into similar pockets consistently achieve lower RMSD. However, the degradation of different models in dissimilar pockets evaluation varies significantly. *Physics-based methods* such as Glide, MOE, and Discovery Studio consistently show a very small gap between similar and dissimilar evaluations, indicating excellent generalizability and outperforming most AI-based approaches in the dissimilar pocket scenario. Earlier *AI docking methods* (e.g., TankBind) and most *AI co-folding methods* (e.g., Chai-1, Protenix, AlphaFold3) suffer steep performance drops—TankBind degrades from 4.79Å to 7.31Å, Chai-1 degrades from 2.36Å to 5.72Å, and Protenix degrades from 2.53Å to 6.30Å—highlighting their overreliance on pocket leakage and lack of adaptability. The latest *AI docking methods*, particularly SurfDock (1.56Å to 2.39Å) and UMD V2 (2.08Å to 3.04Å), demonstrate much smaller gaps and showcase robust generalization.

**Overall Implications.** These analyses collectively suggest that pocket similarity is a key determinant of successful docking, particularly for the cross-docking scenario. *AI co-folding method* and *AI docking methods* reveal a stronger dependence on pocket information, while *physics-based methods* show little sensitivity. Notably, even the state-of-the-art models such as SurfDock and UMD V2 exhibit varying levels of dependence on pocket fidelity, indicating that future improvements in docking may arise from synergistically enhancing both pocket modeling and pose prediction.

### 4.4 IMPACT OF RELAXATION FROM A PHYSICALLY-BASED VALIDATION PERSPECTIVE

We systematically evaluated the docking performance of various methods using the PoseBuster test suite, which comprises 20 physicochemical validation metrics assessing stereochemistry and intra- and intermolecular validity. Figures S8 and S9 illustrate the failure rates of the PB-Valid metric before and after relaxation in self-docking and cross-docking settings, respectively.

**Without Relaxation.** In the absence of relaxation, most *AI docking methods* generate ligand poses that violate physicochemical constraints. Notably, models such as EquiBind, FABind, and DeepDock exhibit a high failure rate in intermolecular validity, especially in the *minimum distance-to-protein* metric, with only approximately 10% of the predictions passing the test. Even SurfDock, which achieves the lowest RMSD, fails in nearly half of its predictions for this metric. Among the *AI docking methods*, UMD V2 demonstrates the best performance on PB-Valid, but still exhibits chirality prediction errors. Among *AI co-folding methods*, NeuralPLexer and RFAA perform poorly with respect to intermolecular validity. AlphaFold3 and similar models show relatively stable performance, but are not immune to chirality errors. In comparison, the recently introduced Boltz-1x model effectively addresses these issues, achieving the highest PB-Valid pass rate among all AI methods. *Physics-based methods* consistently perform well in structural plausibility, achieving high pass rates.

**With Relaxation.** Most AI-based approaches benefit significantly from our relaxation protocol, which effectively mitigates intra- and intermolecular clashes. SurfDock emerges as the top-performing method on the benchmark with post-relaxation. However, relaxation does not resolve chirality errors and UMD V2 shows no performance improvement in this process. Our relaxation module (Appendix C) refines atomic positions and resolves steric clashes (e.g., Figure S13), but it does not correct chirality errors, which involve incorrect stereochemical configurations at tetrahedral centers, requiring specific bond reconfigurations beyond energy minimization. For UMD V2, Figures S8 and S9 show that a significant portion of PB-Valid failures stem from tetrahedral chirality errors. Because relaxation cannot address these stereochemical issues, the PB-Valid scores do not improve. Similarly, *AI co-folding methods*, including AlphaFold3, Chai-1, Boltz-1, and Protenix, exhibit limited

improvement due to persistent chirality errors. For *AI co-folding methods*, tetrahedral chirality failures arise because these models rely on learned patterns from training data, which may not fully capture the precise stereochemical constraints required for correct chiral center configurations. For instance, (Childs et al., 2025) highlights that AlphaFold3 struggles with stereochemical accuracy for D-peptides due to its training data bias toward L-amino acid structures. This limitation extends to other *AI co-folding methods* in our study, except for Boltz-1x, which ensures stereochemical correctness by introducing physics-inspired potentials to fix hallucinations. Figure S11 illustrates two representative cases of chirality errors in docking predictions.

**Summary.** Integrating relaxation with *AI docking methods* yields the state-of-the-art performance. Concurrently, advancements in AI for biology are driving progress in docking methodologies. Boltz-1x incorporates physical mechanisms to produce docking results that satisfy physical constraints without relying on relaxation. These findings highlight the critical role of combining physically informed generation with refinement procedures in docking pipelines, particularly in drug design scenarios that require atomic-level accuracy.

## 5    CONCLUSION

This paper proposed PoseX, a comprehensive benchmark for protein-ligand docking. Specifically, we curated a new dataset with newly released protein-ligand complex crystal structures focusing on both self-docking and cross-docking, and incorporated 23 docking methods across three main research lines (*physics-based methods*, *AI docking methods*, and *AI co-folding methods*) to make an exhaustive comparison. We also designed a novel relaxation module to refine the AI-generated binding pose through energy minimization. Furthermore, we developed an online leaderboard that fosters transparency and facilitates straightforward, fair comparisons in protein-ligand docking. By conducting thorough empirical studies, we drew several key conclusions: (1) Both *AI docking methods* and *AI co-folding methods* have outperformed *physics-based methods* in overall docking success rate. (2) Most structural plausibility (except chirality) of AI-based approaches can be enhanced with relaxation, which means combining AI modeling with physics-based post-processing may achieve excellent performance. (3) Almost all the *AI co-folding methods* are plagued by ligand chirality, except for Boltz-1x, which introduced a new inference time steering technique to fix hallucinations, pointing out the direction of incorporation of advantages of AI and physics. (4) Pocket information can be leveraged adequately, especially for *AI co-folding methods*, to further promote the performance in real-world applications.

## LIMITATION AND FUTURE WORK

Here, we summarize the limitations of this work and present some directions for future research.

1. **Evaluation of binding affinities on downstream tasks.** While we focus on pose prediction and structural plausibility, binding affinity prediction remains an underexplored but complementary objective. Joint evaluation of structure and affinity on downstream tasks, such as drug-target and enzyme-substrate interactions, would enable a more holistic assessment of docking algorithms and remains an exciting direction for future research.

2. **Benchmarking on multi-ligand systems.** So far, most existing benchmarks focus on the evaluation of single-ligand docking, while multi-ligand docking is also practical in real-world applications such as enzyme engineering, where enzymes usually catalyze substrates together with cofactors. Thus, it is worth being assessed exhaustively in the future.

3. **Evaluating DynamicBind with apo structures.** There is an approximation for DynamicBind evaluation where holo structures were employed as input to maintain a uniform input format across the benchmark, rather than the apo structures, which were supposed to be. However, DynamicBind is explicitly designed to handle apo structures and such approximation may constitute an out-of-distribution setting for the model, potentially failing to capture its intended utility or unfairly penalizing its performance. Therefore, we will supplement the evaluation results for apo structures in DynamicBind in the future, making them more rigorous.

## REPRODUCIBILITY STATEMENT

The code used in this paper can be found in https://github.com/CataAI/PoseX. The construction process of PoseX-SD and PoseX-CD, as well as the experiments carried out in this work, could be reproduced by following the instructions in README. The corresponding parameters of all the methods are shown in Appendix B.

## ACKNOWLEDGEMENTS

Tianfan Fu is supported by Young Scientists Fund (C Class) of the National Natural Science Foundation of China (Grant No. 62506154) and the Fundamental Research Funds for the Central Universities and Nanjing University International Collaboration Initiative (Grant No. 020214380129).

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

# A DATASET CONSTRUCTION AND STATISTICAL ANALYSIS

## A.1 DATASET CONSTRUCTION PROCESS

Table S1: Construction process of PoseX Self-Docking (PoseX-SD).

| Selection Step | # proteins (unique PDB IDs) | # ligands (unique CCD IDs) |
|---|---|---|
| PDB entries released from January 1st, 2022 to January 1st, 2025 feature a refinement resolution of 2 Å or better and include at least one protein and one ligand | 13207 | 6877 |
| Remove unknown ligands (e.g., UNX, UNL) | 13202 | 6875 |
| Remove proteins with a sequence length greater than 2000 | 11771 | 6442 |
| Ligands weighing from 100 Da to 900 Da | 9768 | 6196 |
| Ligands with at least 3 heavy atoms | 9706 | 6163 |
| Ligands containing only H, C, O, N, P, S, F, Cl atoms | 9030 | 5741 |
| Ligands that are not covalently bound to protein | 8383 | 5185 |
| Structures with no unknown atoms (e.g., element X) | 8349 | 5166 |
| Ligand real space R-factor is at most 0.2 | 7521 | 4476 |
| Ligand real space correlation coefficient is at least 0.95 | 5734 | 3426 |
| Ligand model completeness is 100% | 5645 | 3358 |
| Ligand starting conformation could be generated with ETKDGv3 | 5638 | 3351 |
| All ligand SDF files can be loaded with RDKit and pass its sanitization | 5634 | 3345 |
| PDB ligand report does not list stereochemical errors | 5600 | 3317 |
| PDB ligand report does not list any atomic clashes | 3971 | 2541 |
| Select single protein-ligand conformation [1] | 3971 | 2541 |
| Intermolecular distance between the ligand(s) and the protein is at least 0.2 Å | 3945 | 2527 |
| Intermolecular distance between ligand(s) and other small organic molecules is at least 0.2 Å | 3889 | 2477 |
| Intermolecular distance between ligand(s) and ion metals in complex is at least 0.2 Å | 3889 | 2477 |
| Remove ligands which are within 5.0 Å of any protein symmetry mate | 2451 | 1598 |
| Get a set with unique pdbs and unique ccds by Hopcroft–Karp matching algorithm | 1587 | 1587 |
| Select representative PDB entries by clustering protein sequences | 718 | 718 |

[1] The first conformation is chosen when multiple conformations are available in the PDB entry.
[2] Clustering with MMseqs2 is done with a sequence identity threshold of 0% and a minimum coverage of 100%.

Table S2: Construction process of PoseX Cross-Docking (PoseX-CD).

| Selection Step | # proteins (unique PDB IDs) | # ligands (unique CCD IDs) |
|---|---|---|
| PDB entries released from January 1st, 2022 to January 1st, 2025 feature a refinement resolution of 2 Å or better and include at least one protein and one ligand | 13207 | 6877 |
| Remove unknown ligands (e.g., UNX, UNL) | 13202 | 6875 |
| Remove proteins with a sequence length greater than 2000 | 11771 | 6442 |
| Ligands weighing from 100 Da to 900 Da | 9768 | 6196 |
| Ligands with at least 3 heavy atoms | 9706 | 6163 |
| Ligands containing only H, C, O, N, P, S, F, Cl atoms | 9030 | 5741 |
| Ligands that are not covalently bound to protein | 8383 | 5185 |
| Structures with no unknown atoms (e.g., element X) | 8349 | 5166 |
| Ligand real space R-factor is at most 0.2 | 7521 | 4476 |
| Ligand real space correlation coefficient is at least 0.95 | 5734 | 3426 |
| Ligand model completeness is 100% | 5645 | 3358 |
| Ligand starting conformation could be generated with ETKDGv3 | 5638 | 3351 |
| All ligand SDF files can be loaded with RDKit and pass its sanitization | 5634 | 3345 |
| PDB ligand report does not list stereochemical errors | 5600 | 3317 |
| PDB ligand report does not list any atomic clashes | 3971 | 2541 |
| Select single protein-ligand conformation [1] | 3971 | 2541 |
| Intermolecular distance between the ligand(s) and the protein is at least 0.2 Å | 3945 | 2527 |
| Intermolecular distance between the ligand(s) and the other ligands is at least 5.0 Å | 2232 | 1536 |
| Remove ligands which are within 5.0 Å of any protein symmetry mate | 1240 | 908 |
| Cluster proteins that have at least 90% sequence identity [2] | 890 | 708 |
| Structures can be successfully aligned to the reference structure in each cluster [3] | 371 | 362 |

[1] The first conformation is chosen when multiple conformations are available in the PDB entry.

[2] Clustering with MMseqs2 is done with a sequence identity threshold of 90% and a minimum coverage of 80%.

[3] Each candidate protein is structurally aligned to the reference protein via the superposition of $C_\alpha$ atom of amino acid residues using PyMOL. A candidate PDB entry is removed if the RMSD of the protein alignment is greater than 2.0 Å and a candidate ligand is removed if it is 4.0 Å away from the reference ligand.

## A.2 STATISTICAL CHARACTERISTICS

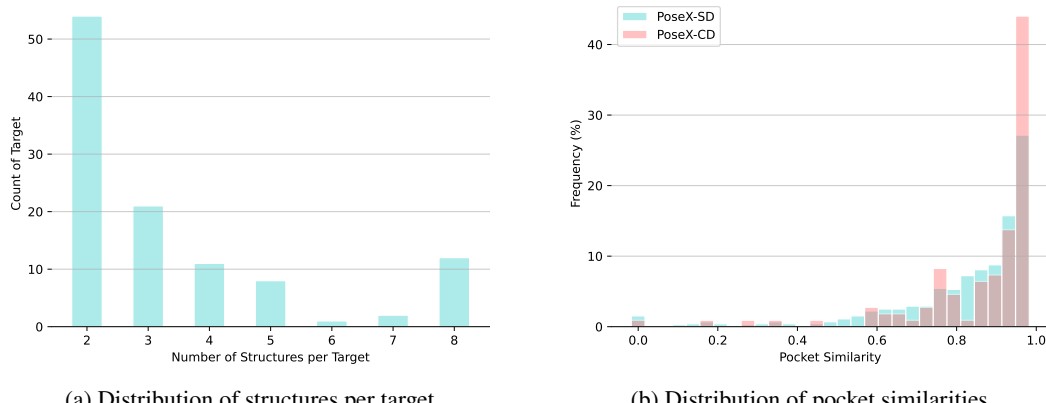

(a) Distribution of structures per target

(b) Distribution of pocket similarities

Figure S1: (a) The distribution of structures per target shows that every protein adopts at least two distinct conformations, and about half of the targets are represented by just two. (b) The distribution of pocket similarities.

## B    DOCKING METHODS AND EVALUATION SETTINGS

This section presents the docking methods employed in our evaluation and illustrates the corresponding setups.

### B.1    PHYSICS-BASED METHODS

*Physics-based methods* employ physical forces and geometric complementarity to model molecular interactions, predicting ligand binding to the target protein. Usually, the atomic coordinates of the protein's binding site remain fixed, while the ligand undergoes flexible conformational changes. This schema reduces the computational complexity of docking simulations by neglecting the protein's dynamic flexibility. Although computationally efficient, this method may fail to fully account for the inherent flexibility of proteins, as biological systems often exhibit conformational changes upon ligand binding. We include 5 *physics-based methods* in this paper, including Discovery Studio (Pawar & Rohane, 2021), Schrödinger Glide (Friesner et al., 2004), MOE (Vilar et al., 2008), AutoDock Vina (Trott & Olson, 2010; Eberhardt et al., 2021) and GNINA (McNutt et al., 2021).

#### B.1.1    SCHRÖDINGER GLIDE

**Schrödinger Glide** is a leading provider of biomolecular simulation software, and Glide is one of its flagship products, focusing on precise molecular docking simulations (Friesner et al., 2004; Bhachoo & Beuming, 2017). Glide adopts a unique hierarchical docking approach, starting with coarse screening and then performing fine optimization on high-scoring results to improve prediction accuracy.

**Software Version**: Schrödinger Suite 2022-1, Build 141

**Docking Workflow**

1. Use **PrepWizard** to preprocess the protein files by adding hydrogens and optimizing with the OPLS3 force field at pH 7.4.

2. Use **LigPrep** to preprocess small molecules, preserving the chirality of the input ligand. Use **Epik** to predict the pKa and protonation states of small molecules at pH 7.0. Optimize the small-molecule conformations using the S-OPLS force field, and output one small-molecule conformation as the input for docking.

3. Define the INNERBOX dimensions as $10 \times 10 \times 10$ Å, and the OUTERBOX dimensions as:

$$\begin{pmatrix} \text{Size}_x \\ \text{Size}_y \\ \text{Size}_z \end{pmatrix} = \begin{pmatrix} x_{\max} - x_{\min} + 20 \\ y_{\max} - y_{\min} + 20 \\ z_{\max} - z_{\min} + 20 \end{pmatrix}$$

   The force field is set to OPLS3, and all other parameters are set by default. Generate a grid file.

4. Perform molecular docking using Glide SP (Standard Precision), and output one small molecule pose as the docking result.

**Runtime Environment**: Run on an Intel i9-10920X CPU using 16 cores.

#### B.1.2    DISCOVERY STUDIO

**Discovery Studio** (Pawar & Rohane, 2021), developed by Dassault Systèmes BIOVIA, is a comprehensive life sciences research platform that covers molecular modeling, virtual screening, and more. For protein-ligand binding, Discovery Studio performs conformational sampling around a given binding site and ranks potential poses using physics-based scoring functions like CDOCKER (which combines grid-based molecular dynamics and CHARMM force fields).

**Software Version**: v2021.1.0.20298.

**Docking Workflow**:

1. Use the **Proteins Preparation** components in Discovery Studio to process the protein files. The protein was protonated at pH 7.4 with a solvent ionic strength of 0.145 M. Minimization was performed using the **CHARMm** force field to optimize the protein structure, and all other parameters are set by default.

2. Use the **Ligands Preparation** components in Discovery Studio to process the ligand files. Enumerate ionization states for each ligand within a pH range of 6.5-8.5. Enumerate automeric forms for each ligand with a maximum of 10 tautomers per ligand. Fix the bad valencies by adjusting formal charges, and all other parameters are set by default.

3. Dock the prepared proteins and the corresponding prepared ligands using the **CDOCKER** components in Discovery Studio. The docking site was centered at:

$$\begin{pmatrix} x_c \\ y_c \\ z_c \end{pmatrix} = \begin{pmatrix} \frac{x_{\max}+x_{\min}}{2} \\ \frac{y_{\max}+y_{\min}}{2} \\ \frac{z_{\max}+z_{\min}}{2} \end{pmatrix}$$

Define the binding sphere radius as:

$$R = \max\{(x_{\max} - x_{\min}), (y_{\max} - y_{\min}) - (z_{\max} - z_{\min})\} + 20$$

The docking simulations were performed using the **CHARMm** force field. Assign the partial charges to the ligands via the **Momany-Rone** method, and all other parameters are set by default. 10 top docking poses output each docking run, and the best-scored pose was selected as the final docking result.

**Runtime Environment**: Run on an Intel Ultra 5 125H CPU using 14 cores.

### B.1.3  MOLECULAR OPERATING ENVIRONMENT (MOE)

**Molecular Operating Environment (MOE)** (Vilar et al., 2008), developed by the Canadian company Chemical Computing Group, is a commercial drug discovery software platform that combines visualization, modeling, simulations, and methodology development into a single, unified package.

**Software Version**: MOE 2024.06.

**Docking Workflow**

1. An SVL script automates the docking pipeline.

2. The **StructurePreparation** function is employed to preprocess protein structures.

3. The binding site is defined by reference ligands.

4. The **Triangle Matcher** algorithm is utilized to generate initial ligand poses.

5. The scoring function is configured as **London dG**, with a maximum of 30 poses generated.

6. Poses are refined using a fixed receptor, optimizing only the ligand's position and conformation, with the re-scoring function configured as **GBVI/WSA dG** and a maximum of 5 poses retained.

**Runtime Environment**: Run on an AMD EPYC 9554 CPU.

### B.1.4  AUTODOCK VINA

**AutoDock Vina** (Eberhardt et al., 2021) is one of the fastest and most widely used **open-source** molecule docking programs. It combines global search (to identify potential binding modes) with local optimization (to refine these modes).

**Software Versions**

- AutoDock-Vina: 1.2.6
- MGLTools: 1.5.7
- Reduce: 4.14.230914
- OpenBabel: 3.1.0

- Meeko: 0.6.1

**Docking Workflow**

1. Use **Reduce** to add polar hydrogens to the protein structure.

2. Use **OpenBabel** to add non-polar hydrogens and normalize atom names, exporting the protein in a format recognizable by MGLTools.

3. Use the **receptor_prepare4.py** script from MGLTools to convert the hydrogen-added protein PDB file into a PDBQT file.

4. Use **OpenBabel** to add hydrogens to the ligand molecule at pH 7.4.

5. Use the **mk_prepare_ligand.py** script from Meeko to convert the hydrogen-added ligand SDF file into a PDBQT file.

6. Define the docking box center and size as follows:

$$\begin{pmatrix} x_c \\ y_c \\ z_c \end{pmatrix} = \begin{pmatrix} \frac{x_{\max}+x_{\min}}{2} \\ \frac{y_{\max}+y_{\min}}{2} \\ \frac{z_{\max}+z_{\min}}{2} \end{pmatrix}$$

$$\begin{pmatrix} \text{Size}_x \\ \text{Size}_y \\ \text{Size}_z \end{pmatrix} = \begin{pmatrix} x_{\max} - x_{\min} + 20 \\ y_{\max} - y_{\min} + 20 \\ z_{\max} - z_{\min} + 20 \end{pmatrix}$$

7. Perform molecular docking using the prepared protein and ligand PDBQT files.

8. Use **vina_split** to split the output file, extract the best-scored pose for each ligand, and convert the resulting PDBQT file into an SDF file using Meeko for the final output.

**Runtime Environment**: Run on an AMD EPYC 9554 CPU, with no specified core limit and up to 256 cores available.

### B.1.5 GNINA

**GNINA** (McNutt et al., 2021; 2025) is a relatively new project that introduces DL techniques into the field of molecular docking, particularly leveraging convolutional neural networks (CNNs) as scoring functions to improve docking scoring. It is an **open-source** software.

**Docker Image**: https://hub.docker.com/layers/gnina/gnina/latest/images

**Running Parameters**: The command used is:

```
gnina -r rec.pdb -l lig.sdf -autobox_ref.sdf -o out.sdf,
```

where `lig.sdf` is `PDB_CCD_ligand_start_conf.sdf` and `ref.sdf` is `PDB_CCD_ligand.sdf`.

**Runtime Environment**: Run on Nvidia A6000 GPU.

### B.2 AI DOCKING METHODS

*AI docking methods* utilize SMILES strings of ligands and three-dimensional structures of protein targets as input to predict energetically favorable ligand conformations bound to target proteins. These methods systematically explore the conformational space of small molecules to identify low-energy configurations that optimize the binding affinity to proteins. By sampling diverse ligand conformations, *AI docking methods* enhance the optimization of spatial arrangements to maximize interactions with protein active sites, including hydrogen bonds, hydrophobic interactions, and electrostatic complementarity. We involve 11 *AI docking methods* in this paper, including DeepDock (Méndez-Lucio et al., 2021), EquiBind (Stärk et al., 2022), TankBind (Lu et al., 2022), DiffDock (Corso et al., 2022), UMD V2 (Alcaide et al., 2024), FABind (Pei et al., 2023), DiffDock-L (Corso et al., 2024), DiffDock-Pocket (Plainer et al., 2023), DynamicBind (Lu et al., 2024), Interformer (Lai et al., 2024) and SurfDock (Cao et al., 2024).

### B.2.1 DEEPDOCK

**DeepDock** (Méndez-Lucio et al., 2021) is a geometric DL model that learns a statistical potential based on the distance likelihood.

**GitHub Repository**: `https://github.com/OptiMaL-PSE-Lab/DeepDock`

**GitHub Commit Hash**: ab1e45044c5e0a69105b48d09ea984c6a5ebc26c

**Running Parameters**: Default parameters are used in evaluation.

**Runtime Environment**: Run on Intel(R) Xeon(R) CPU E5-2620 v4.

### B.2.2 EQUIBIND

**EquiBind** (Stärk et al., 2022) is an SE(3)-equivariant geometric DL model designed for direct-shot prediction of both i) the receptor binding site (blind docking) and ii) the ligand's bound pose and orientation.

**GitHub Repository**: `https://github.com/HannesStark/EquiBind`

**GitHub Commit Hash**: 41bd00fd6801b95d2cf6c4d300cd76ae5e6dab5e

**Running Parameters**: Default parameters are used in evaluation.

**Runtime Environment**: Run on Nvidia A6000 GPU.

### B.2.3 TANKBIND

**TankBind** (Lu et al., 2022) incorporates trigonometric constraints as a robust inductive bias into the model, and explicitly examines all potential binding sites for each protein by dividing the entire protein into functional blocks. establishes an efficient diffusion process within this space.

**GitHub Repository**: `https://github.com/luwei0917/TankBind`

**GitHub Commit Hash**: ff85f511db11d7a3e648d2e01cd6fdb4f9823483

**Running Parameters**: Use the structure of the entire protein as input for prediction, rather than chains within 10Å of the ligand in the default setting.

**Runtime Environment**: Run on an AMD EPYC 9554 CPU.

### B.2.4 DIFFDOCK

**DiffDock** (Corso et al., 2022) is a diffusion-based generative model defined on the non-Euclidean manifold of ligand poses. It maps this manifold to the product space of the degrees of freedom (translational, rotational, and torsional) relevant to docking and establishes an efficient diffusion process within this space.

**GitHub Repository**: `https://github.com/gcorso/DiffDock`

**GitHub Commit Hash**: bc6b5151457ea5304ee69779d92de0fded599a2c

**Running Parameters**: Default parameters are used in evaluation.

**Runtime Environment**: Run on Nvidia A800 GPU.

### B.2.5 DIFFDOCK-L

**DiffDock-L** (Corso et al., 2024) is a variant of DiffDock that scales up data and model size by integrating synthetic data strategies.

**GitHub Repository**: `https://github.com/gcorso/DiffDock`

**GitHub Commit Hash**: b4704d94de74d8cb2acbe7ec84ad234c09e78009

**Running Parameters**: `samples_per_complex` is changed from the default value of 10 to 40.

**Runtime Environment**: Run on Nvidia A800 GPU.

### B.2.6 DIFFDOCK-POCKET

**DiffDock-Pocket** (Plainer et al., 2023) is a variant of DiffDock with additional binding pocket specification.

**GitHub Repository**: `https://github.com/plainerman/DiffDock-Pocket`

**GitHub Commit Hash**: 3902bdd4d42ee5254d37aa694d005a992c92ad93

**Running Parameters**: Default parameters are used in evaluation.

**Runtime Environment**: Run on Nvidia A6000 GPU.

### B.2.7 DYNAMICBIND

**DynamicBind** (Lu et al., 2024) utilizes equivariant geometric diffusion networks to generate a smooth energy landscape, facilitating efficient transitions between various equilibrium states. DynamicBind accurately identifies ligand-specific conformations from unbound protein structures, eliminating the need for holo-structures or extensive sampling.

**GitHub Repository**: `https://github.com/luwei0917/DynamicBind`

**GitHub Commit Hash**: abdcd83f313cd20d50c3917e04615e989a8f63e5

**Running Parameters**: Input proteins are holo structures provided by the PoseX dataset, and internal relaxation is disabled.

**Runtime Environment**: Run on Nvidia A800 GPU.

### B.2.8 FABIND

**FABind** (Pei et al., 2023) is an end-to-end model that integrates pocket prediction and docking to achieve precise and efficient protein-ligand binding predictions. It involves a ligand-informed pocket prediction module, which is also utilized to enhance the accuracy of docking pose estimation.

**GitHub Repository**: `https://github.com/QizhiPei/FABind`

**GitHub Commit Hash**: bc6b5151457ea5304ee69779d92de0fded599a2c

**Running Parameters**: Default parameters are used in evaluation.

**Runtime Environment**: Run on Nvidia A800 GPU.

### B.2.9 UNI-MOL DOCKING V2

**UMD V2** (Alcaide et al., 2024) represents Uni-Mol Docking v2. It combines the pretrained molecular and pocket models to learn the distance matrix and then uses a coordinate model to predict the molecule's final coordinates.

**GitHub Repository**: `https://github.com/deepmodeling/Uni-Mol/tree/main/unimol_docking_v2`

**GitHub Commit Hash**: c0365df6535b90197246399417a9b21250268352

**Running Parameters**: Default parameters are used in prediction. About one-fifth of the molecules in the model output will encounter RDKit's sanitization check errors. This issue is resolved by reading in the correct molecular topology and then assigning the coordinates predicted by **Uni-Mol Docking v2** to the molecules with the new topology

**Runtime Environment**: Run on Nvidia A6000 GPU.

### B.2.10 INTERFORMER

**Interformer** (Lai et al., 2024), a unified model based on the Graph-Transformer architecture, is specifically designed to capture non-covalent interactions using an interaction-aware mixture density network. Furthermore, it employs a negative sampling strategy to adjust the interaction distribution, thereby improving affinity prediction accuracy.

**GitHub Repository**: `https://github.com/tencent-ailab/Interformer`

**GitHub Commit Hash**: 8cced9b8a5d8c887787a8c8731d9f087563d4c7e

**Running Parameters**: Use `PDB_CCD_ligand.sdf` to obtain the pocket, perform UFF optimization on `PDB_CCD_ligand_start_conf.sdf` and replace it in the `uff` folder, and use the `-uff_as_ligand` option during prediction.

**Runtime Environment**: Run on Nvidia A6000 GPU.

### B.2.11 SURFDOCK

**SurfDock** (Cao et al., 2024) combines protein sequences, three-dimensional structural graphs, and surface-level features within an equivariant architecture. It leverages a generative diffusion model on a non-Euclidean manifold to optimize molecular translations, rotations, and torsions, producing accurate and reliable binding poses.

**GitHub Repository**: `https://github.com/CAODH/SurfDock`

**GitHub Commit Hash**: 2f0422f6ddcfdfefc3fa61ef12a1d6406a589bce

**Running Parameters**: Default parameters are used in evaluation.

**Runtime Environment**: Run on Nvidia A6000 GPU.

### B.3 AI CO-FOLDING METHODS

*AI co-folding methods* represent a significant advance in computational biology by simultaneously predicting the conformation of both the protein and its associated ligand, which sets them apart from *physics-based methods* and *AI docking methods*. In contrast to *physics-based methods*, which typically assume a fixed protein structure and focus on optimizing ligand placement, or *AI docking methods* that may still rely on predefined protein conformations, *AI co-folding methods* adopt a more holistic strategy–**taking only the protein's amino acid sequence and ligand's SMILES strings as input**. These methods aim to capture the dynamic interaction between proteins and ligands by predicting their structures in tandem, enabling a more accurate representation of how these molecules interact in biological systems. In this paper, we involve 7 *AI co-folding methods*, including NeuralPLexer (Qiao et al.), RoseTTAFold-All-Atom (RFAA) (Krishna et al., 2024), AlphaFold3 (Abramson et al., 2024), Chai-1 (Discovery et al., 2024), Boltz-1 (Wohlwend et al., 2024), Boltz-1x (Wohlwend et al., 2024) and Protenix (Team et al., 2025). It should be noted that in our evaluation of *AI co-folding methods*, we did not consider post-translational modifications and used unmodified protein sequences as input.

### B.3.1 NEURALPLEXER

**NeuralPLexer** (Qiao et al.) is a physics-inspired flow-based generative model for biomolecular complex structure prediction based on sequences only. NeuralPLexer combines a protein language model with graph encoding to learn sequence information and represent 3D molecular structures and bioactivity information.

**GitHub Repository**:`https://github.com/zrqiao/NeuralPLexer`

**GitHub Commit Hash**: 2c52b10d3094e836661dfecfa3be76f47dcdea7e

**Running Parameters**: Default parameters are used in evaluation.

**Runtime Environment**: Run on Nvidia A6000 GPU.

### B.3.2 ROSETTAFOLD-ALL-ATOM

**RoseTTAFold-All-Atom (RFAA)** (Krishna et al., 2024) is a generalized foundation model for all-atom biomolecular structure prediction and design, including protein, nucleic acid, and other small molecules. RoseTTAFold-All-Atom is a 3-track based architecture incorporating equivariant neural networks for all atomic structure prediction. Meanwhile, it integrates with RFDiffusion for molecular design.

**GitHub Repository**: `https://github.com/baker-laboratory/RoseTTAFold-All-Atom`

**GitHub Commit Hash**: 6c8514053acf76da0f9edde2aa51b40abff68fa1

**Running Parameters**: Default parameters are used in evaluation.

**Runtime Environment**: Run on Nvidia A800 GPU.

### B.3.3 ALPHAFOLD3

**AlphaFold3** (Abramson et al., 2024), developed by DeepMind, represents the latest advancement in protein structure prediction technology. Building on the successes of its predecessor AlphaFold 2 (Jumper et al., 2021)), AlphaFold3 adopts a diffusion model instead of a structure module in AlphaFold2, not only improving the accuracy of protein folding but also supporting the structure prediction of complexes (e.g., protein-RNA, protein-ligand), which enables its usage in protein-ligand docking.

**Software Version**: 3.0.0

**Running Parameters**: Except for the number of seeds being set to 1, the rest of the predictions are made using the default parameters. We finally select the top 1 result for evaluation.

**Runtime Environment**: Run on Nvidia A800 GPU.

### B.3.4 CHAI-1

**Chai-1** (Boitreaud et al., 2024) is a multimodal molecular foundation model that can also predict structures with a single sequence. By leveraging the decoder-only Transformer framework, which is widely used in Large Language Models (LLM) like GPT, Chai-1 encodes sequential information without database search. Moreover, Chai-1 accepts various chemical or biological constraint features as input to predict more accurate molecular structures.

**Software Version**: 0.5.2

**Running Parameters**: Use the online MSA server to obtain MSA information, keep the rest as default settings, and select the top 1 result for evaluation.

**Runtime Environment**: Run on Nvidia A800 GPU.

### B.3.5 BOLTZ-1

**Boltz-1** (Wohlwend et al., 2024) aims to reproduce AlphaFold3 and release all code (model architecture, training, and inference), achieving competitive performance. Additionally, Boltz-1 introduces several architectural innovations, including a novel reverse diffusion process and a revamped confidence model, enhancing its predictive accuracy and robustness.

**Software Version**: 0.4.0

**Running Parameters**: Use the MSA online server to obtain MSA information, set diffusion samples to 5, and select the top 1 result for evaluation.

**Runtime Environment**: Run on Nvidia A800 GPU.

### B.3.6 BOLTZ-1x

**Boltz-1x** (Wohlwend et al., 2024) is an advanced version of the Boltz-1 model. It introduces a novel inference-time steering technique, which enhances the physical quality of predicted poses by reducing hallucinations and non-physical predictions. This ensures more reliable and biologically plausible structures.

**Software Version**: 1.0.0

**Running Parameters**: Use the MSA online server to obtain MSA information, set diffusion samples to 5, and select the top 1 result for evaluation.

**Runtime Environment**: Run on Nvidia A800 GPU.

### B.3.7 PROTENIX

**Protenix** (Team et al., 2025) is a comprehensive and open-source reproduction of AlphaFold3, developed by ByteDance. It introduces several architectural innovations, including a modular PyTorch framework that facilitates full training and inference, and optimizations such as custom CUDA kernels and BF16 training to enhance computational efficiency.

**Software Version**: 0.4.2

**Running Parameters**: Use the MSA online server to obtain MSA information, and the seed is set to 101.

**Runtime Environment**: Run on Nvidia A6000 GPU.

## B.4 Training Data Cutoff Times

Table S3: Training Data Cutoff Times for Different Methods

| Method | Training Data Cutoff Time |
|---|---|
| **Traditional physics-based methods** | |
| Discovery Studio (Pawar & Rohane, 2021) | N/A |
| Schrödinger Glide (Friesner et al., 2004) | N/A |
| MOE (Vilar et al., 2008) | N/A |
| AutoDock Vina (Trott & Olson, 2010; Eberhardt et al., 2021) | N/A |
| GNINA (McNutt et al., 2021) | 2018-12 |
| **AI docking methods** | |
| DeepDock (Méndez-Lucio et al., 2021) | 2018-12 |
| EquiBind (Stärk et al., 2022) | 2018-12 |
| TankBind (Lu et al., 2022) | 2018-12 |
| DiffDock (Corso et al., 2022) | 2018-12 |
| UMD V2 (Alcaide et al., 2024) | 2018-12 |
| FABind (Pei et al., 2023) | 2018-12 |
| DiffDock-L (Corso et al., 2024) | 2018-12 |
| DiffDock-Pocket (Plainer et al., 2023) | 2018-12 |
| DynamicBind (Lu et al., 2024) | 2018-12 |
| Interformer (Lai et al., 2024) | 2018-12 |
| SurfDock (Cao et al., 2024) | 2018-12 |
| **AI co-folding methods** | |
| NeuralPLexer (Qiao et al.) | 2018-12 |
| RoseTTAFold-All-Atom (Krishna et al., 2024) | 2021-11 |
| AlphaFold3 (Abramson et al., 2024) | 2021-10 |
| Chai-1 (Discovery et al., 2024) | 2021-02 |
| Boltz-1 (Wohlwend et al., 2024) | 2021-10 |
| Boltz-1x (Wohlwend et al., 2024) | 2021-10 |
| Protenix (Team et al., 2025) | 2021-10 |

## C   TECHNICAL DETAILS OF RELAXATION PROCESS

Our relaxation is based on the following software: OpenMM 7.7 (Eastman et al., 2017), PDBFixer 1.8 (Eastman et al., 2012-2025), RDKit 2023.09 (Contributors, 2006-2024), AmberTools 23, and OpenFF 2.1.0 (Consortium, 2024). It contains the following essential steps:

- Structure preprocessing and integrity restoration. Use PDBFixer (v1.8) to handle the initial structure files:
  - Parse complete protein sequence information from CIF files, retaining water molecules and metal ions within a 5 Å range of the ligand in AI-predicted models.
  - Standardize non-canonical amino acids to canonical forms (e.g., SEP to SER), simultaneously correcting the protein sequence database.
  - Detect structural deficiencies using the findMissingResidues/findMissingAtoms algorithms, and apply the AddMissingAtoms module to complete atoms (including N-terminal ACE and C-terminal NME capping).
- Molecular topology construction and validation. To address the lack of bond order information in PDBFixer:
  - Integrate Amber ff14SB force field atom types and topology bond parameters to establish bond order matching rules.
  - Build a molecular graph model with RDKit (v2023.09) and perform SanitizeMol standardization checks (including charge correction and stereochemistry validation).
  - Apply the RDKit AddHs module for protonation, optimizing the spatial arrangement of hydrogen atoms.
- Force field parameterization. Employ a multi-scale force field combination strategy:
  - For protein systems: Generate Amber ff14SB force field parameters using OpenMM 7.7.
  - For ligand systems: Perform OpenFF 2.1.0 (Consortium, 2024) parameterization using the OpenFF 2.1.0 toolkit, including mmff94s charge calculations and XML topology generation.
- Constrained molecular dynamics optimization. Implement energy minimization on the OpenMM 7.7 platform (Eastman et al., 2017):
  - Constraints: Apply additional forces $(0.5 * k * ((x - x_0)^2 + (y - y_0)^2 + (z - z_0)^2)$ (where $k = 10$, $x_0, y_0, z_0$ are original 3D coordinates) to constrain backbone atomic positions in the protein structure, keeping newly added atoms free.
  - Integration parameters: Langevin thermostat (300 K, friction coefficient 1 $ps^{-1}$), time step 0.004 ps.
  - Convergence criteria: Energy gradient convergence threshold $\leq$ 10 kJ/mol/nm.

# D  DESCRIPTION OF VALIDITY

The validity checks for the structures analyzed in this study were conducted using PoseBuster (Buttenschoen et al., 2024), a tool for assessing the reliability and accuracy of molecular poses. The validation process encompasses chemical validity and consistency, intramolecular validity, and intermolecular validity, each assessed with specific criteria as detailed below. In this study, we define **structural plausibility** as stereochemical correctness and intra- and intermolecular validity.

## D.1  CHEMICAL VALIDITY AND CONSISTENCY

- **File loads**: The input molecule can be successfully loaded into a molecule object by RDKit.
- **Sanitisation**: The input molecule passes RDKit's chemical sanitisation checks, ensuring it adheres to basic chemical rules.
- **Molecular formula**: The molecular formula of the input molecule is identical to that of the true molecule.
- **Bonds**: The bonds in the input molecule are the same as in the true molecule.
- **Tetrahedral chirality**: The specified tetrahedral chirality in the input molecule is the same as in the true molecule.
- **Double bond stereochemistry**: The specified double bond stereochemistry in the input molecule is the same as in the true molecule.

## D.2  INTRAMOLECULAR VALIDITY

- **Bond lengths**: The bond lengths in the input molecule are within 0.75 of the lower and 1.25 of the upper bounds determined by distance geometry.
- **Bond angles**: The angles in the input molecule are within 0.75 of the lower and 1.25 of the upper bounds determined by distance geometry.
- **Planar aromatic rings**: All atoms in aromatic rings with 5 or 6 members are within 0.25 Å of the closest shared plane.
- **Planar double bonds**: The two carbons of aromatic carbon-carbon double bonds and their ring neighbours are within 0.25 Å of the closest shared plane.
- **Internal steric clash**: The interatomic distance between pairs of non-covalently bound atoms is above 0.7 of the lower bound determined by distance geometry.
- **Energy ratio**: The calculated energy of the input molecule is no more than 100 times the average energy of an ensemble of 50 conformations generated for the input molecule. The energy is calculated using the UFF in RDKit, and the conformations are generated with ETKDGv3, followed by force field relaxation using the UFF with up to 200 iterations.

## D.3  INTERMOLECULAR VALIDITY

- **Minimum protein-ligand distance**: The distance between protein-ligand atom pairs is larger than 0.75 times the sum of the pairs' van der Waals radii.
- **Minimum distance to organic cofactors**: The distance between ligand and organic cofactor atoms is larger than 0.75 times the sum of the pair's van der Waals radii.
- **Minimum distance to inorganic cofactors**: The distance between ligand and inorganic cofactor atoms is larger than 0.75 times the sum of the pair's covalent radii.
- **Volume overlap with protein**: The share of ligand volume that intersects with the protein is less than 7.5%. The volumes are defined by the van der Waals radii around the heavy atoms, scaled by 0.8.
- **Volume overlap with organic cofactors**: The share of ligand volume that intersects with organic cofactors is less than 7.5%. The volumes are defined by the van der Waals radii around the heavy atoms scaled by 0.8.
- **Volume overlap with inorganic cofactors**: The share of ligand volume that intersects with inorganic cofactors is less than 7.5%. The volumes are defined by the van der Waals radii around the heavy atoms scaled by 0.5.

# E  ADDITIONAL TABLES AND FIGURES FOR MODEL EVALUATION

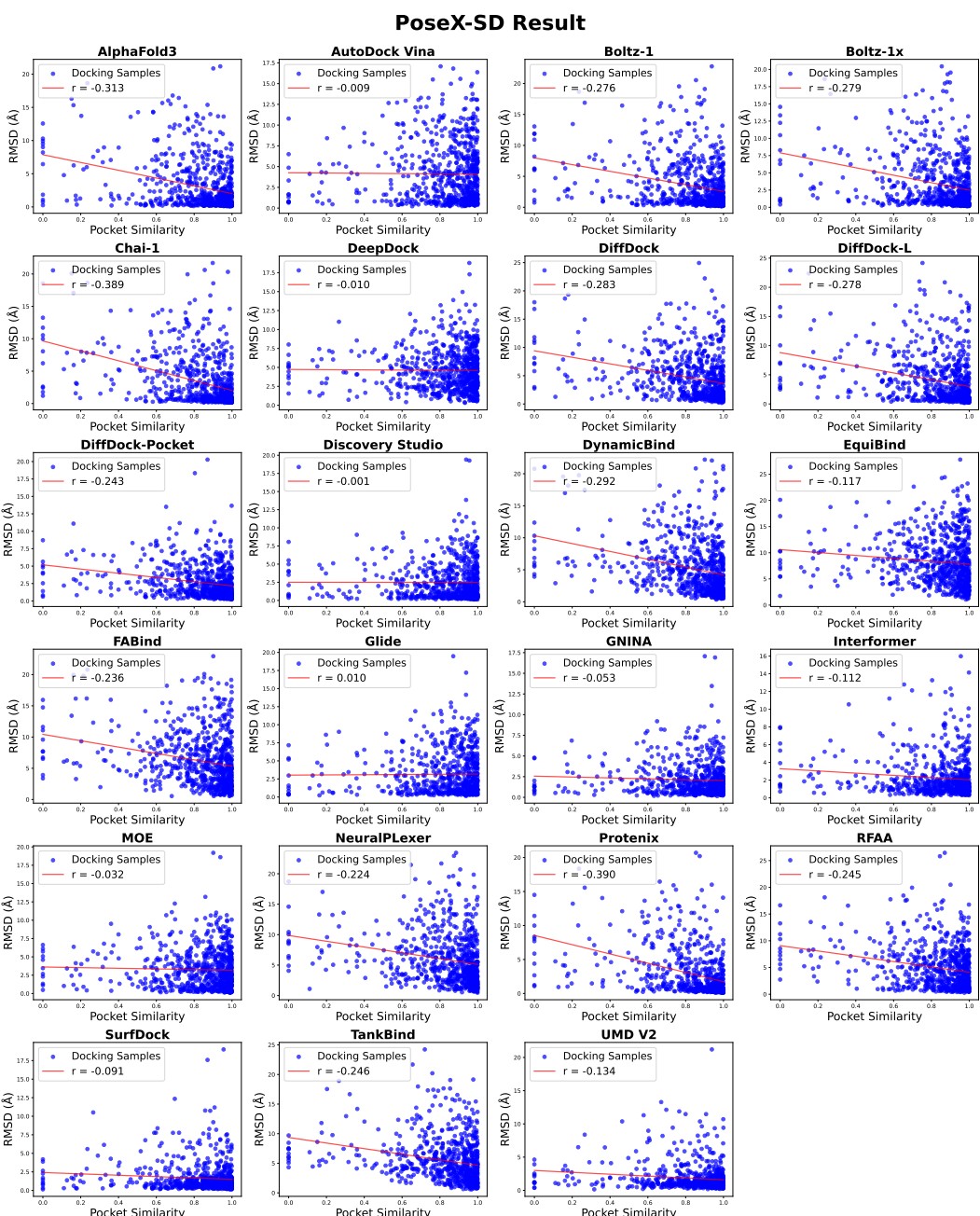

Figure S2: The performance of most AI-based approaches is significantly influenced by pocket similarity under **self-docking** setup. Among them, Protenix (Team et al., 2025) exhibits the strongest negative correlation (r = -0.390), whereas SurfDock (Cao et al., 2024), an AI-based model, demonstrates minimal statistical association. In contrast, *physics-based methods*, such as AutoDock Vina and Glide, are relatively unaffected by protein similarity.

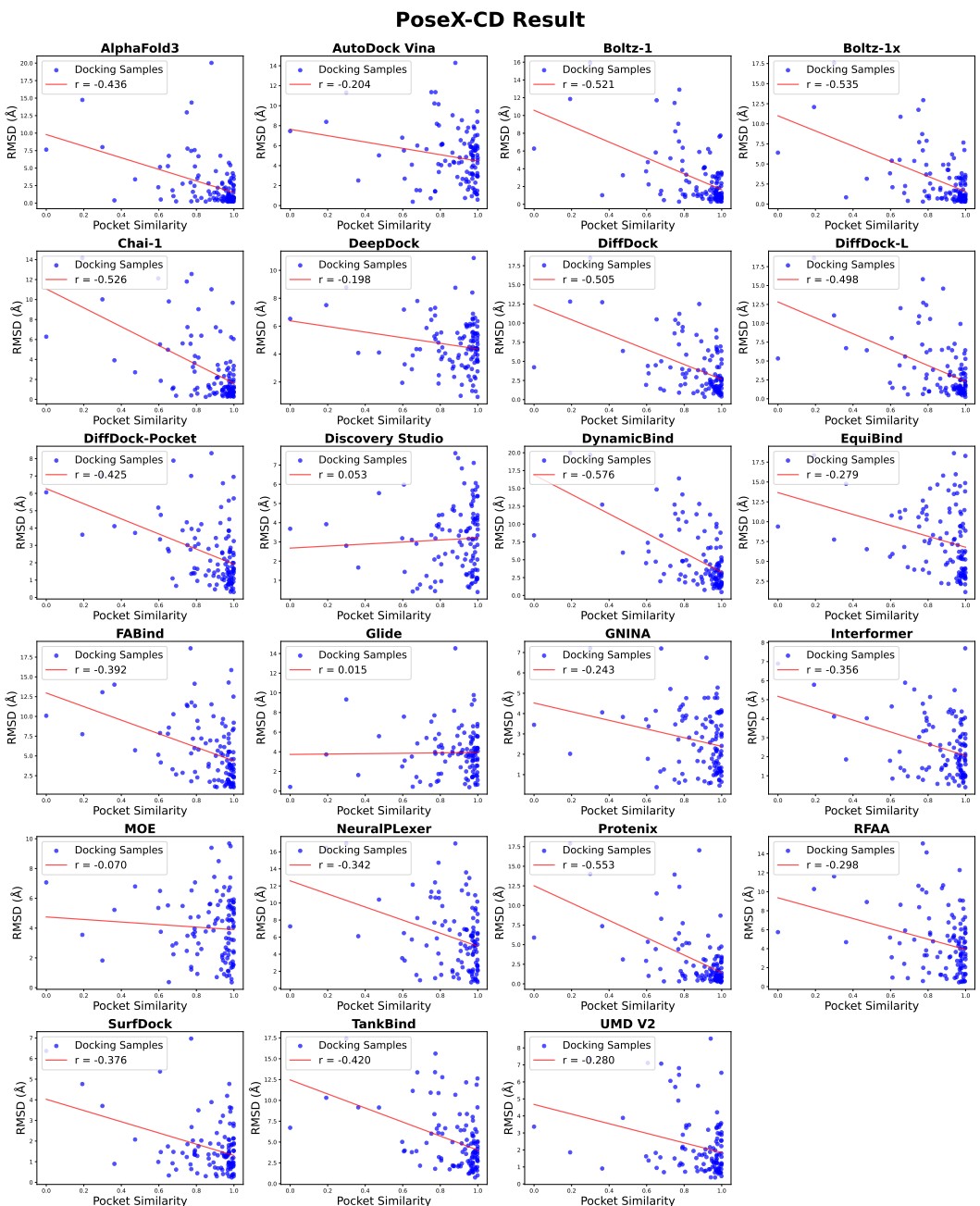

Figure S3: The performance of most AI-based approaches is significantly influenced by the pocket similarity in **cross-docking** scenario, where similar conclusions as the self-docking scenario can be derived.

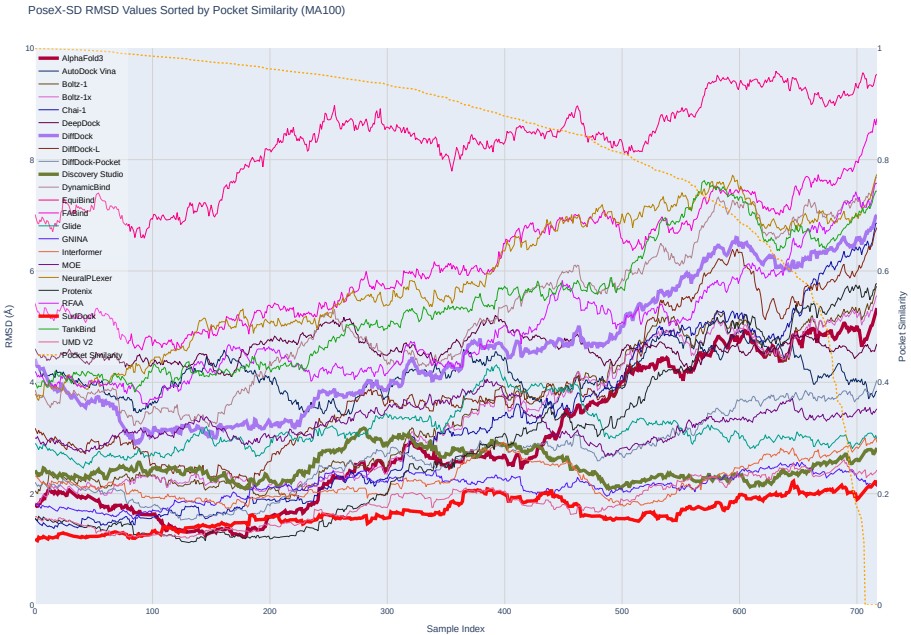

Figure S4: **Performance on the PoseX-SD dataset sorted by pocket similarity.** Samples are sorted by pocket similarity in descending order, and the RMSD results are processed with a moving average (window size: 100). It can be seen that most AI-based approaches degrade as pocket similarity decreases, while *physics-based methods* perform relatively stably.

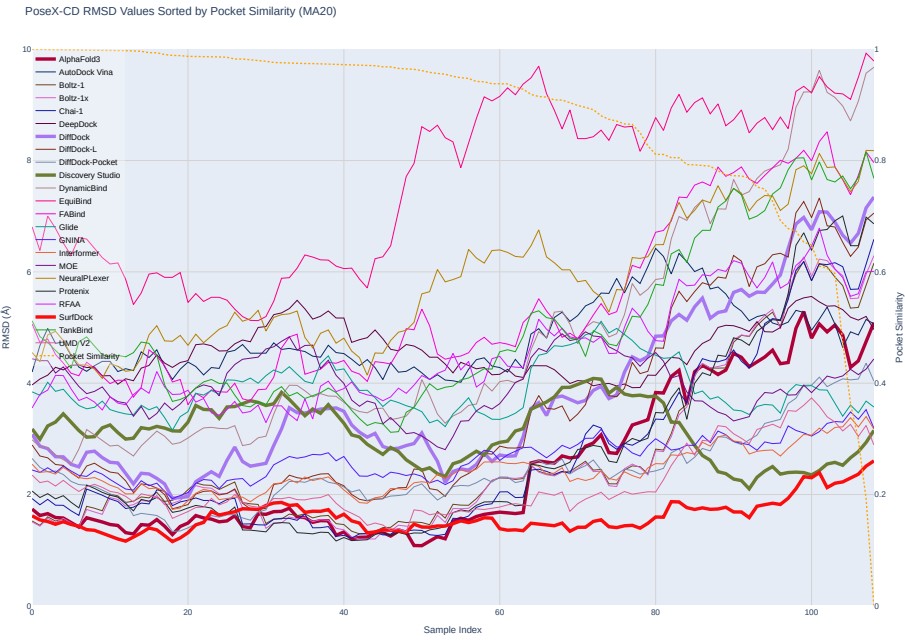

Figure S5: **Performance on the PoseX-CD dataset sorted by pocket similarity.** The results are similar to those on PoseX-SD (window size: 20).

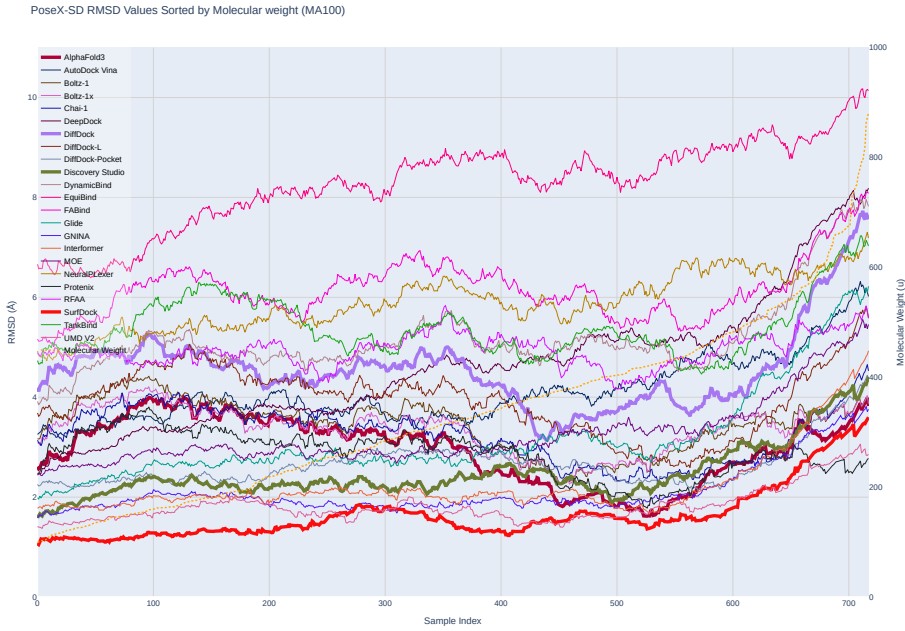

Figure S6: **Performance on the PoseX-SD dataset sorted by molecular weight.** Samples are sorted by molecular weight in ascending order, and the RMSD results are processed with a moving average (window size: 100). It can be seen that most approaches exhibit a clear performance drop for ligands heavier than 450 Da.

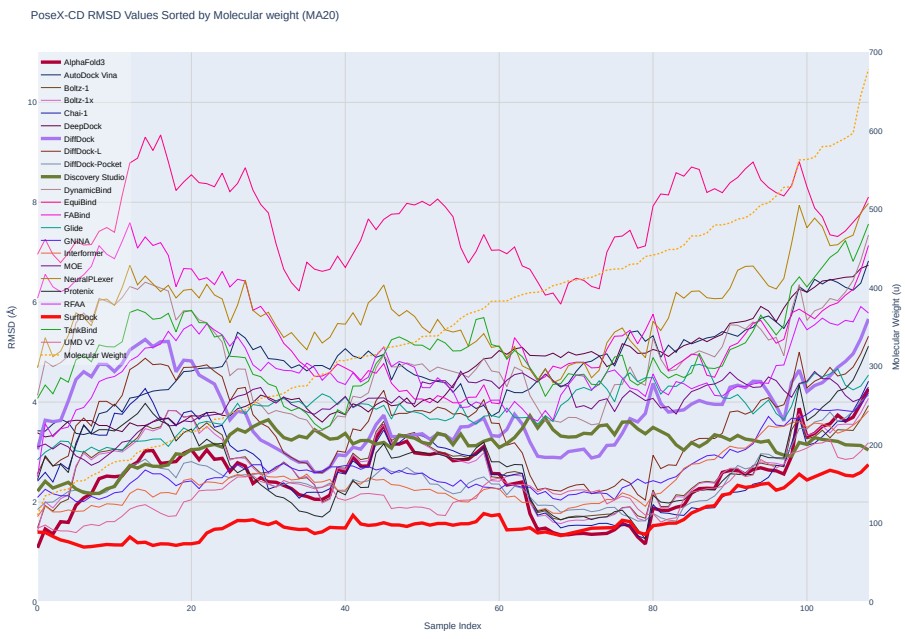

Figure S7: **Performance on the PoseX-CD dataset sorted by molecular weight.** The results are similar to those on PoseX-SD (window size: 20).

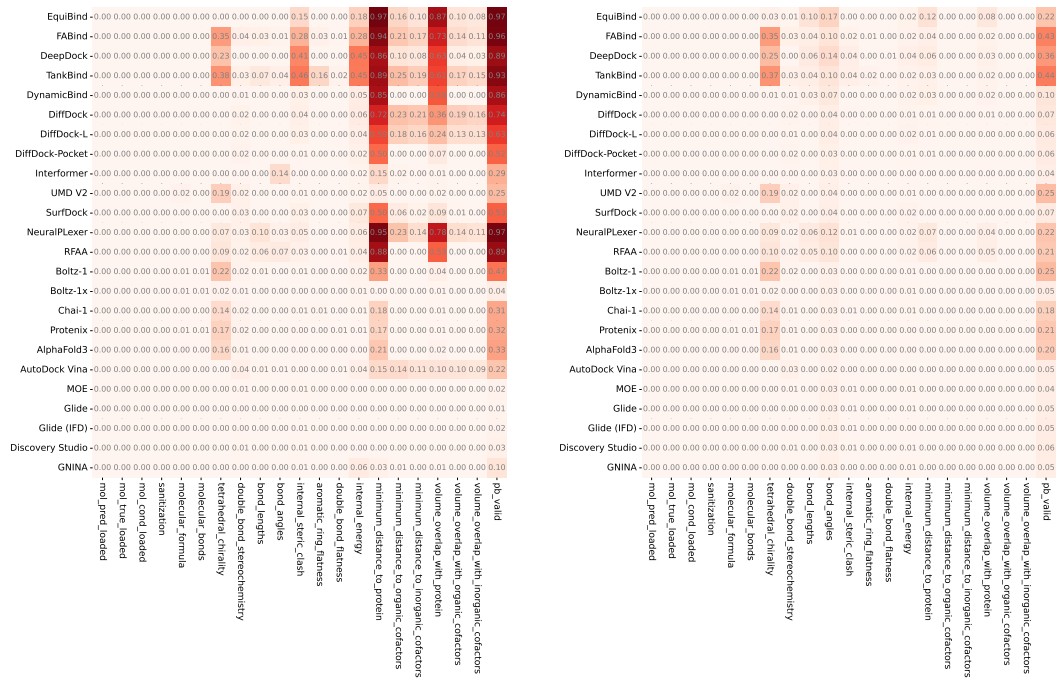

Figure S8: The proportion of models filtered out based on various filtering criteria in PB-Valid (PoseX-SD).

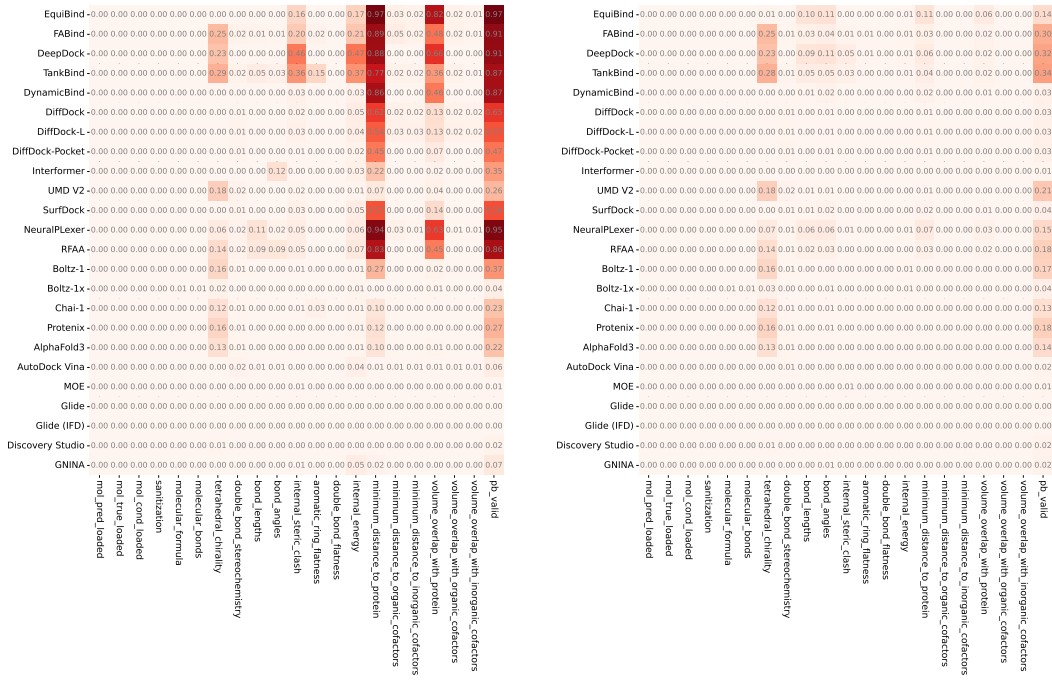

Figure S9: The proportion of models filtered out based on various filtering criteria in PB-Valid (PoseX-CD).

Table S4: Performance (success rate) on PoseX-SD, PoseX-CD and Astex

| Methods | PoseX-SD RMSD < 2Å w/o relax. | PoseX-SD RMSD < 2Å w/ relax. | PoseX-SD RMSD < 2Å & PB-Valid w/o relax. | PoseX-SD RMSD < 2Å & PB-Valid w/ relax. | PoseX-CD RMSD < 2Å w/o relax. | PoseX-CD RMSD < 2Å w/ relax. | PoseX-CD RMSD < 2Å & PB-Valid w/o relax. | PoseX-CD RMSD < 2Å & PB-Valid w/ relax. | Astex RMSD < 2Å w/o relax. | Astex RMSD < 2Å w/ relax. | Astex RMSD < 2Å & PB-Valid w/o relax. | Astex RMSD < 2Å & PB-Valid w/ relax. |
|---|---|---|---|---|---|---|---|---|---|---|---|---|
| SurfDock | 77.07±0.73 | 78.04±0.52 | 41.73±0.13 | 73.67±0.79 | 76.00±0.66 | 77.04±0.03 | 37.06±1.78 | 74.36±0.02 | 91.76±0.69 | 94.07±0.25 | 62.16±0.88 | 89.35±0.37 |
| UMD V2 | 73.68±0.00 | 72.42±0.00 | 60.58±0.00 | 59.19±0.00 | 69.48±0.00 | 69.24±0.00 | 57.35±0.00 | 58.25±0.00 | 94.12±0.00 | 94.12±0.00 | 85.88±0.00 | 84.71±0.00 |
| Interformer | 65.50±0.66 | 66.58±0.79 | 48.89±0.99 | 64.07±0.99 | 58.55±0.65 | 60.19±0.64 | 42.47±0.41 | 59.85±0.88 | 79.87±0.65 | 83.77±0.71 | 60.26±0.67 | 81.09±0.93 |
| DiffDock-Pocket | 52.83±0.69 | 52.65±0.59 | 29.39±0.30 | 50.28±0.63 | 58.02±0.38 | 58.53±0.77 | 34.16±0.43 | 57.21±1.06 | 83.60±0.52 | 84.90±0.67 | 60.18±0.36 | 83.56±0.83 |
| DiffDock-L | 46.57±1.05 | 47.12±0.87 | 25.39±0.56 | 44.80±0.97 | 53.38±1.48 | 54.69±0.69 | 29.39±0.90 | 53.17±0.73 | 86.08±1.25 | 86.26±0.77 | 73.23±0.71 | 84.88±0.84 |
| DiffDock | 36.07±0.20 | 36.81±0.77 | 16.72±0.99 | 35.10±0.80 | 45.42±1.01 | 47.15±1.13 | 18.83±0.88 | 46.07±1.27 | 76.66±0.57 | 76.13±0.93 | 60.86±0.93 | 73.76±1.01 |
| DynamicBind | 24.64±0.61 | 27.00±0.73 | 7.98±0.59 | 25.77±0.79 | 30.39±0.92 | 32.77±0.85 | 9.50±0.72 | 32.06±1.00 | 62.30±0.75 | 66.09±0.78 | 19.90±0.65 | 59.02±0.89 |
| TankBind | 16.62±0.17 | 19.64±0.20 | 3.81±0.35 | 12.76±0.17 | 20.21±0.47 | 25.84±0.67 | 4.30±0.15 | 16.34±0.47 | 56.59±0.31 | 57.69±0.41 | 5.92±0.24 | 32.91±0.31 |
| DeepDock | 16.39±0.33 | 19.13±0.56 | 6.13±0.30 | 15.69±0.82 | 17.13±0.29 | 20.21±0.91 | 4.64±0.38 | 16.36±1.19 | 30.54±0.31 | 34.03±0.72 | 10.50±0.34 | 23.63±0.99 |
| FABind | 15.32±0.23 | 17.87±0.67 | 2.28±0.07 | 12.81±0.11 | 24.72±0.20 | 29.38±0.41 | 4.00±0.02 | 21.15±0.58 | 45.83±0.21 | 44.58±0.53 | 9.41±0.04 | 30.52±0.32 |
| EquiBind | 3.48±0.00 | 4.46±0.00 | 0.42±0.00 | 4.32±0.00 | 5.11±0.00 | 6.56±0.00 | 0.40±0.00 | 6.56±0.00 | 10.59±0.00 | 11.76±0.00 | 2.35±0.00 | 9.41±0.00 |
| AlphaFold3 | 60.65±0.18 | 60.31±0.31 | 45.28±0.36 | 51.61±0.43 | 68.87±0.61 | 68.79±0.41 | 53.79±0.81 | 62.88±0.33 | 83.47±0.38 | 82.29±0.36 | 75.34±0.56 | 76.47±0.38 |
| Protenix | 56.69±0.00 | 56.27±0.39 | 44.20±0.72 | 47.40±0.33 | 61.56±0.63 | 61.09±0.45 | 47.96±0.98 | 49.84±0.05 | 82.40±0.29 | 81.19±0.42 | 66.94±0.84 | 71.73±0.18 |
| Chai-1 | 56.41±0.59 | 56.08±0.33 | 44.10±0.53 | 49.58±0.00 | 67.06±0.68 | 67.02±0.91 | 55.12±1.27 | 60.36±0.99 | 82.31±0.63 | 82.57±0.59 | 70.80±0.87 | 74.21±0.45 |
| Boltz-1 | 54.22±1.19 | 53.71±1.02 | 35.70±0.93 | 42.76±0.49 | 65.11±1.16 | 64.11±0.77 | 41.78±0.80 | 51.94±0.52 | 69.13±1.17 | 69.52±0.88 | 52.98±0.86 | 57.67±0.50 |
| Boltz-1x | 53.99±0.07 | 53.71±0.24 | 52.60±0.17 | 50.98±0.69 | 64.82±0.76 | 64.39±0.60 | 61.44±0.95 | 61.19±1.01 | 71.78±0.38 | 73.02±0.40 | 70.56±0.52 | 71.64±0.84 |
| Boltz-2 [1] | 65.32±0.10 | 65.44±0.29 | 51.32±0.28 | 55.54±0.38 | 73.43±0.57 | 72.63±0.35 | 61.73±0.75 | 64.99±0.18 | 77.73±0.31 | 77.54±0.32 | 64.71±0.49 | 67.19±0.27 |
| RFAA | 30.25±0.58 | 31.49±0.42 | 6.94±0.78 | 26.73±0.73 | 31.95±0.82 | 32.56±0.69 | 10.89±1.15 | 29.70±1.12 | 37.33±0.69 | 36.22±0.54 | 9.75±0.95 | 32.70±0.91 |
| NeuralPLexer | 14.30±0.52 | 17.92±0.33 | 1.58±0.26 | 15.37±0.33 | 22.47±0.47 | 25.61±0.21 | 1.51±0.56 | 21.90±0.06 | 28.46±0.49 | 35.25±0.26 | 0.00±0.00 | 30.66±0.18 |
| GNINA | 64.58±0.69 | 64.44±0.65 | 60.49±0.72 | 61.79±0.46 | 53.10±0.80 | 54.13±1.42 | 49.93±0.95 | 53.19±1.35 | 81.22±0.74 | 81.36±1.00 | 81.19±0.82 | 80.81±0.86 |
| Discovery Studio | 54.74±0.00 | 54.87±0.00 | 54.04±0.00 | 52.65±0.00 | 44.08±0.00 | 43.68±0.00 | 43.28±0.00 | 42.89±0.00 | 67.06±0.00 | 67.06±0.00 | 65.88±0.00 | 64.71±0.00 |
| Glide | 48.33±0.00 | 47.91±0.00 | 47.91±0.00 | 46.24±0.00 | 37.45±0.00 | 38.44±0.00 | 36.99±0.00 | 38.44±0.00 | 63.53±0.00 | 64.71±0.00 | 63.53±0.00 | 63.53±0.00 |
| Glide (IFD) | 46.24±0.00 | 46.52±0.00 | 46.10±0.00 | 45.40±0.00 | 44.80±0.00 | 44.95±0.00 | 44.65±0.00 | 44.65±0.00 | 67.06±0.00 | 67.06±0.00 | 67.06±0.00 | 67.06±0.00 |
| AutoDock Vina | 40.01±0.48 | 39.89±0.51 | 36.92±0.52 | 38.27±0.33 | 28.08±0.65 | 28.67±0.93 | 27.79±0.72 | 28.13±0.87 | 56.27±0.56 | 56.39±0.70 | 52.06±0.61 | 55.10±0.58 |
| MOE | 40.25±0.00 | 39.42±0.00 | 39.55±0.00 | 37.74±0.00 | 33.28±0.00 | 33.33±0.00 | 33.27±0.00 | 33.17±0.00 | 56.47±0.00 | 57.65±0.00 | 56.47±0.00 | 57.65±0.00 |

[1] We additionally evaluated Boltz-2, a recently released and highly influential model. However, the training dataset for Boltz-2 was constructed from PDB structures available as of June 1, 2023, which overlaps with our evaluation dataset.

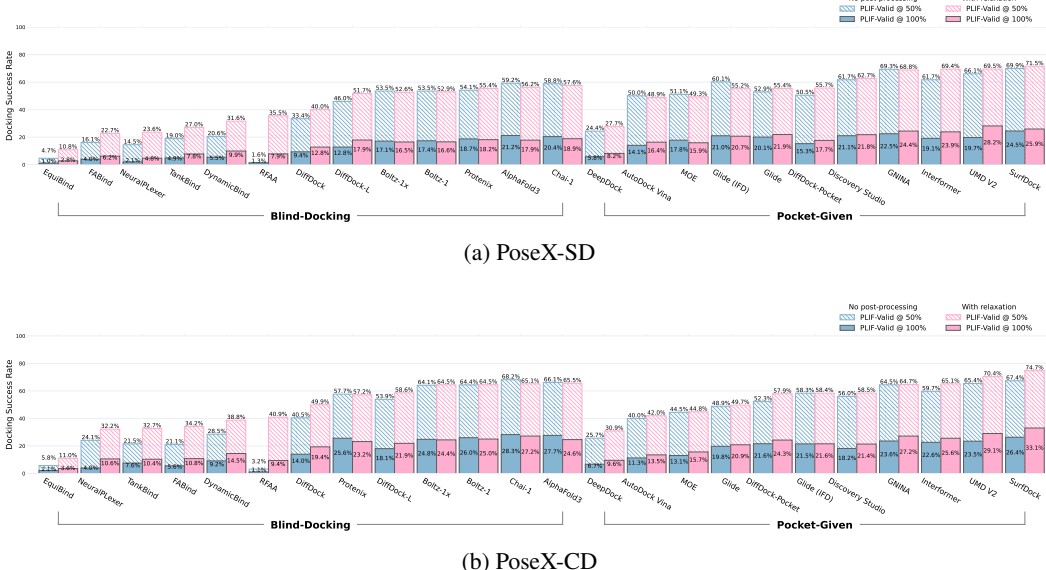

(a) PoseX-SD

(b) PoseX-CD

Figure S10: **PLIF-Valid** (Errington et al., 2025) evaluation on PoseX-SD and PoseX-CD for all methods. PLIF-Valid assesses recovery rates of protein-ligand interaction fingerprints, which identify the protein residue, the interaction type, and, optionally, the ligand atom involved in the interaction. Overall, the ranking of methods on PLIF-Valid closely follows the docking success rate ranking in Figure 2 and Figure 3, with SurfDock achieving the best performance. The relaxation module consistently improves PLIF-Valid for nearly all methods. Notably, physics-based methods show better interaction preservation than their geometric success rates suggest (e.g., PoseX-SD: DiffDock-L success rate 47.1% > Glide (IFD) 46.5%, but PLIF-Valid 51.7% < Glide (IFD) 55.2%; PoseX-CD: DiffDock success rate 47.1% > Glide IFD 44.8%, but PLIF-Valid 49.9% < Glide IFD 58.8%), likely due to their implicit consideration of protein–ligand interactions. The detailed results with standard deviation are reported in Table S5.

Table S5: Performance (PLIF-Valid) on PoseX-SD, PoseX-CD and Astex

| Methods | PoseX-SD PLIF-valid @ 50% w/o relax. | w/ relax. | PLIF-valid @ 100% w/o relax. | w/ relax. | PoseX-CD PLIF-valid @ 50% w/o relax. | w/ relax. | PLIF-valid @ 100% w/o relax. | w/ relax. | Astex PLIF-valid @ 50% w/o relax. | w/ relax. | PLIF-valid @ 100% w/o relax. | w/ relax. |
|---|---|---|---|---|---|---|---|---|---|---|---|---|
| SurfDock | 69.89±0.54 | 71.52±0.79 | 24.49±0.25 | 25.94±0.61 | 67.37±0.42 | 74.71±0.32 | 26.35±0.58 | 33.06±0.53 | 77.91±0.47 | 70.52±0.53 | 26.04±0.40 | 29.79±0.57 |
| UMD V2 | 66.08±0.00 | 69.49±0.00 | 19.74±0.00 | 28.18±0.00 | 65.43±0.00 | 70.44±0.00 | 23.48±0.00 | 29.06±0.00 | 70.73±0.00 | 69.51±0.00 | 19.51±0.00 | 30.49±0.00 |
| Interformer | 61.71±0.81 | 69.42±1.49 | 19.15±0.96 | 23.87±1.21 | 59.72±0.83 | 65.12±0.39 | 22.65±1.06 | 25.62±0.94 | 66.47±0.82 | 69.38±0.89 | 26.53±1.01 | 29.92±1.06 |
| DiffDock-Pocket | 50.50±0.46 | 55.73±0.64 | 15.29±0.23 | 17.67±0.46 | 52.28±0.34 | 57.90±0.32 | 21.63±0.50 | 24.30±0.57 | 75.72±0.39 | 72.08±0.47 | 33.03±0.35 | 29.41±0.51 |
| DiffDock-L | 46.01±0.33 | 51.69±0.30 | 12.82±0.05 | 17.94±0.54 | 53.88±0.58 | 58.58±1.44 | 18.05±0.66 | 21.92±0.45 | 72.07±0.44 | 66.89±0.82 | 29.36±0.33 | 29.76±0.49 |
| DiffDock | 33.43±0.55 | 40.02±1.38 | 9.36±0.30 | 12.79±1.02 | 40.45±1.56 | 49.90±0.73 | 14.00±1.36 | 19.38±0.73 | 61.26±1.01 | 63.69±1.02 | 22.16±0.78 | 25.85±0.86 |
| DynamicBind | 20.60±0.79 | 31.64±1.48 | 5.45±0.64 | 9.94±1.10 | 28.54±0.79 | 38.77±0.37 | 9.17±1.00 | 14.53±0.96 | 47.13±0.79 | 51.46±0.87 | 7.63±0.80 | 18.57±1.02 |
| TankBind | 18.96±0.72 | 27.01±1.07 | 4.87±0.30 | 7.82±0.37 | 21.50±0.53 | 32.71±0.46 | 7.59±0.15 | 10.39±0.29 | 39.68±0.62 | 48.35±0.74 | 8.70±0.22 | 17.37±0.33 |
| DeepDock | 24.43±0.46 | 27.71±0.79 | 5.79±0.28 | 8.21±0.61 | 25.69±0.39 | 30.89±0.28 | 6.73±0.53 | 9.58±0.48 | 30.98±0.42 | 35.08±0.51 | 7.52±0.39 | 9.79±0.54 |
| FABind | 16.09±0.29 | 22.73±0.47 | 3.96±0.41 | 6.19±0.18 | 21.05±0.13 | 34.16±0.95 | 5.56±0.21 | 10.80±0.50 | 36.31±0.20 | 46.53±0.69 | 15.08±0.30 | 18.38±0.33 |
| EquiBind | 4.68±0.00 | 10.85±0.00 | 1.02±0.00 | 2.79±0.00 | 5.83±0.00 | 10.97±0.00 | 2.12±0.00 | 3.57±0.00 | 3.66±0.00 | 9.64±0.00 | 0.00±0.00 | 2.41±0.00 |
| AlphaFold3 | 59.21±0.43 | 56.22±1.82 | 21.20±0.83 | 17.86±0.44 | 66.12±0.80 | 65.53±1.08 | 27.68±0.76 | 24.64±0.57 | 75.80±0.60 | 77.61±1.42 | 32.27±0.79 | 29.25±0.50 |
| Protenix | 54.10±0.25 | 55.41±1.06 | 18.74±0.56 | 18.20±0.26 | 57.68±0.49 | 57.19±0.50 | 25.63±0.48 | 23.20±0.40 | 73.85±0.36 | 70.21±0.75 | 32.64±0.52 | 36.34±0.32 |
| Chai-1 | 58.79±0.42 | 57.56±1.58 | 20.41±0.25 | 18.88±0.28 | 68.16±0.02 | 65.13±0.56 | 28.27±0.39 | 27.22±0.21 | 74.42±0.20 | 69.90±1.02 | 30.86±0.31 | 30.45±0.24 |
| Boltz-1 | 53.52±0.36 | 52.91±1.64 | 17.37±0.94 | 16.61±0.09 | 64.36±0.38 | 64.52±1.13 | 26.00±0.91 | 25.03±0.74 | 76.05±0.37 | 72.87±1.36 | 29.36±0.92 | 23.86±0.38 |
| Boltz-1x | 53.54±0.18 | 52.64±0.41 | 17.15±0.22 | 16.51±0.30 | 64.11±0.88 | 64.45±0.73 | 24.83±1.23 | 24.43±0.96 | 77.06±0.50 | 78.36±0.56 | 27.11±0.68 | 30.93±0.60 |
| Boltz-2 | 62.41±0.48 | 61.00±2.05 | 23.06±0.93 | 20.27±0.44 | 71.77±0.87 | 67.97±1.24 | 31.36±0.81 | 27.75±0.66 | 74.54±0.66 | 73.52±1.61 | 34.86±0.86 | 25.79±0.54 |
| RFAA | 1.65±0.39 | 35.48±1.04 | 1.27±0.56 | 7.87±0.36 | 3.19±0.53 | 40.90±0.54 | 1.11±0.52 | 9.41±0.66 | 2.47±0.45 | 38.96±0.77 | 0.00±0.00 | 11.39±0.50 |
| NeuralPLexer | 14.51±0.43 | 23.60±1.28 | 2.06±0.56 | 4.76±0.29 | 24.08±0.54 | 32.24±0.62 | 3.96±0.43 | 10.60±0.48 | 26.06±0.48 | 35.19±0.92 | 3.83±0.49 | 8.53±0.38 |
| GNINA | 69.31±0.59 | 68.83±0.95 | 22.49±0.45 | 24.44±0.64 | 64.46±0.56 | 64.69±0.72 | 23.60±0.73 | 27.20±0.88 | 70.16±0.57 | 66.90±0.82 | 23.91±0.58 | 29.84±0.75 |
| Discovery Studio | 61.70±0.00 | 62.66±0.00 | 21.05±0.00 | 21.82±0.00 | 56.04±0.00 | 58.49±0.00 | 18.20±0.00 | 21.40±0.00 | 60.49±0.00 | 60.98±0.00 | 23.46±0.00 | 23.17±0.00 |
| Glide | 52.86±0.00 | 55.41±0.00 | 20.06±0.00 | 21.93±0.00 | 48.88±0.00 | 49.71±0.00 | 19.76±0.00 | 20.88±0.00 | 62.20±0.00 | 62.20±0.00 | 18.29±0.00 | 25.61±0.00 |
| Glide (IFD) | 60.15±0.00 | 55.25±0.00 | 21.02±0.00 | 20.70±0.00 | 58.33±0.00 | 58.40±0.00 | 21.52±0.00 | 21.58±0.00 | 61.73±0.00 | 58.54±0.00 | 13.58±0.00 | 18.29±0.00 |
| AutoDock Vina | 50.03±0.36 | 48.88±0.45 | 14.07±0.20 | 16.38±0.28 | 39.95±0.21 | 41.97±0.24 | 11.30±0.33 | 13.53±0.43 | 47.64±0.28 | 59.13±0.34 | 17.14±0.26 | 21.79±0.35 |
| MOE | 51.09±0.00 | 49.27±0.00 | 17.81±0.00 | 15.94±0.00 | 44.46±0.00 | 44.81±0.00 | 13.06±0.00 | 15.67±0.00 | 53.09±0.00 | 50.00±0.00 | 24.69±0.00 | 21.95±0.00 |

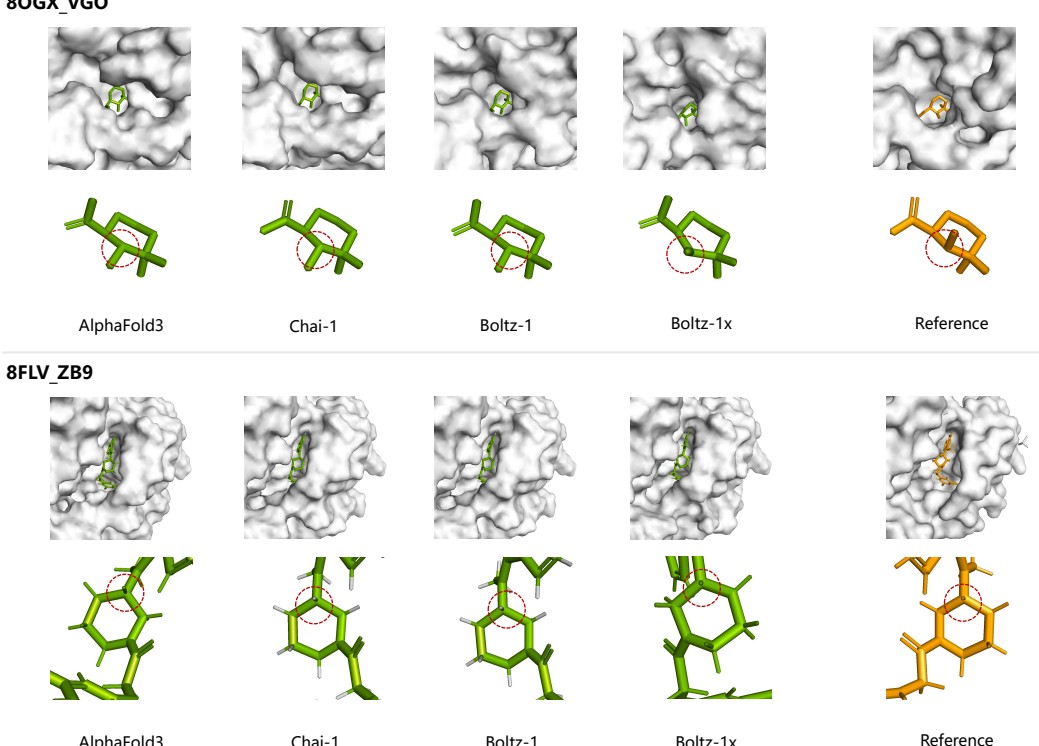

Figure S11: **Case study of *AI co-folding methods* in chirality validation.** We compared AlphaFold3, Chai-1, Boltz-1, and Boltz-1x models on the **8OGX_VGO** and **8FLV_ZB9** complexes. The figure illustrates the docking results, with chiral centers marked by red circles, revealing that all co-folding models except Boltz-1x exhibit chirality errors.

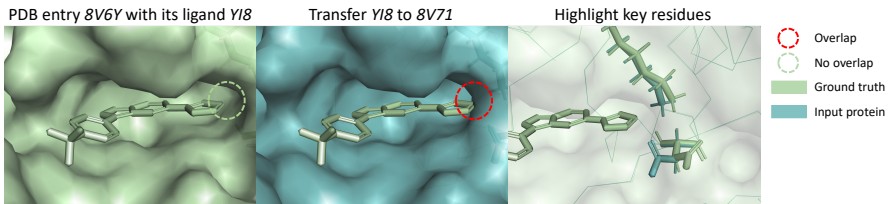

Figure S12: Analysis of **8V71_YI8** in PoseX-CD. When transferring the ligand from its co-crystal structure to the protein structure used for docking through structural alignment, steric clashes arise between the ligand and the protein, underscoring the challenges associated with cross-docking. In this case, all *physics-based methods* failed (RMSD ≥ 2Å), while the top-performing *AI docking method* and *AI co-folding method* (SurfDock and AlphaFold3, respectively) accurately predicted the pose. The rightmost column illustrates the conformational variations in residues that overlap with the ligand across the two protein structures.

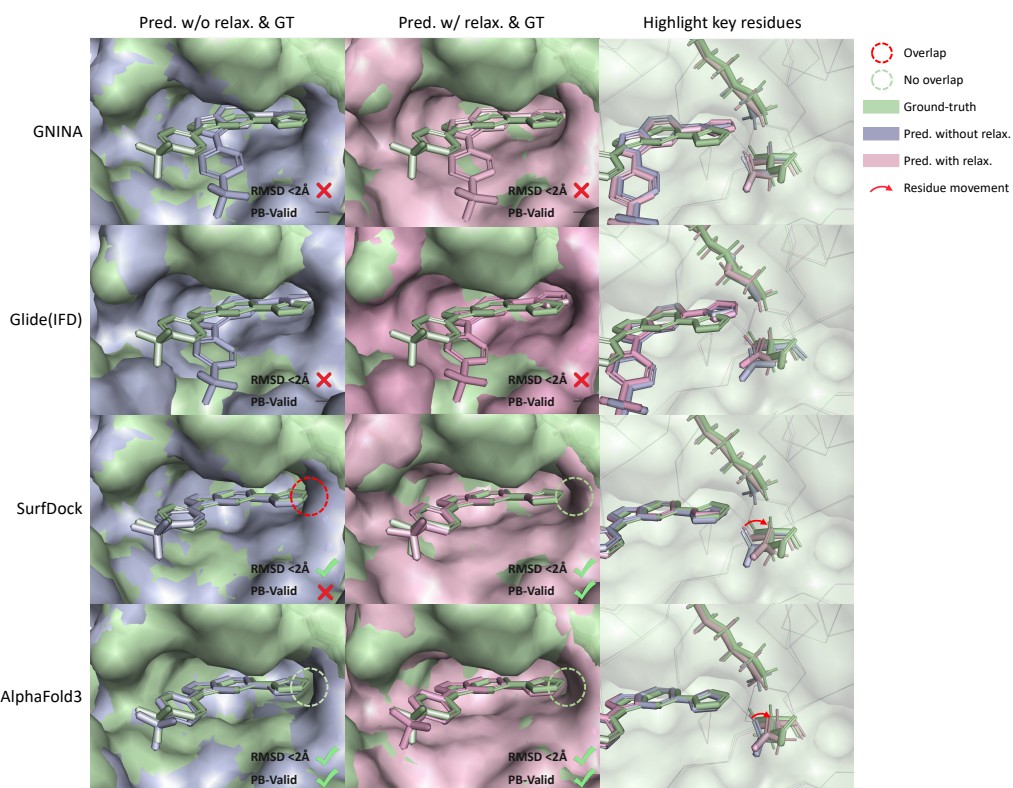

Figure S13: Analysis of docking results for **8V71_YI8**. The *physics-based methods* GNINA and Glide (IFD) generate ligand conformations that substantially deviate from the ground-truth structure. In contrast, SurfDock and AlphaFold3 generate docking poses that closely align with the ground-truth structure. SurfDock's docking poses exhibit steric clashes, which are resolved through relaxation, whereas AlphaFold3's poses are sterically compatible. The rightmost column demonstrates that, for both SurfDock and AlphaFold3, key residues shift toward their corresponding positions in the ground-truth structure after relaxation.

