# OpenReview forum: "PoseX: AI Defeats Physics-based Methods on Protein Ligand Cross-Docking"
_ICLR.cc/2026/Conference — ICLR 2026 Poster_

### Official Review · Reviewer_yfAx · 2025-10-21

**Soundness:** 3
**Presentation:** 2
**Contribution:** 2
**Rating:** 4
**Confidence:** 4

**Summary:**

The authors introduce PoseX, a deep learning benchmark for protein-ligand self and cross-docking. The benchmark is well built and comprehensive in terms of method selection, and empirical results match up to expectations. Nonetheless, some concerns remain regarding the novelty and presentation of these results in the context of a machine learning conference.

**Strengths:**

1. The authors' proposed benchmark is comprehensive and introduces results for protein-ligand cross-docking.
2. Metrics reported for the benchmark are standardized and informative.
3. The results of the benchmark match expectations overall, offering a few subtle insights, e.g., about how to improve ligand chirality prediction.
4. The presence of an online leaderboard is a plus.

**Weaknesses:**

1. The benchmark, at first glance, reads like an expanded version of PoseBench extended to protein-ligand cross-docking. For example, in Line 149, the authors highlight post-prediction relaxation as a contribution of their work, though a version of this technique was already heavily explored in PoseBench (with support for multi-ligands as well).
2. Introducing cross-docking seems like a marginal change over previous works. It is simply a matter of selecting different protein conformations. From a biological perspective, it is interesting to study, but from a machine learning perspective, it is practically identical to what has been done before.
3. The way the authors present their benchmarks' results (as well constructed as they are) is ideal for a scientific journal but less common for a machine learning venue. For example, with the font size and scale of Figure 2, it is hard to quickly discern which machine learning method performs "best" in the self and cross-docking contexts (with all caveats in mind). Maybe highlighting the "most notable" results with a box around them would be a way to address this? Also, the methods in Figure 2 are unorganized (i.e., without category labels), so it is hard for a machine learning person (without expertise) to know which method is a model based purely on deep learning and which is a physics-based method.
4. Why did the authors select Astex Diverse as a benchmark dataset instead of the PoseBusters Benchmark dataset (which is arguably more well-known now and less prone to show up in each method's training set)?

**Questions:**

1. How extensible is PoseX's codebase? For example, how hard is it for current users to build on top of the benchmark to add new datasets or prediction tasks for evaluation? Do you have any case studies in this direction?

---

> ### Author Response · Authors · 2025-11-21
>
> >Weakness 1: The benchmark, at first glance, reads like an expanded version of PoseBench extended to protein-ligand cross-docking. For example, in Line 149, the authors highlight post-prediction relaxation as a contribution of their work, though a version of this technique was already heavily explored in PoseBench (with support for multi-ligands as well).
>
> Thanks for reviewing! We do not think PoseX is an expanded version of PoseBench, since PoseBench focus on self-docking evaluation with AlphaFold3 predicted apo structure, while we mainly focus on a different aspect of cross-docking evaluation on real crystal protein-ligand complex structure, which is more practical in drug development. As for the relaxation module, compared to PoseBench, we have optimized the process such as: (1) we retain water molecules and metal ions within a 5 Å range of the ligand; (2) we address the lack of bond order information in PDBFixer (please refer to Appendix C for details). We consider that all of these are more likely to be an engineering-level or tool-level contribution. Furthermore, the relaxation module employed in PoseBench is also the one developed in DynamicBind, just application for energy minimization, while the technique itself is not heavily explored in PoseBench.
>
> >Weakness 2: Introducing cross-docking seems like a marginal change over previous works. It is simply a matter of selecting different protein conformations. From a biological perspective, it is interesting to study, but from a machine learning perspective, it is practically identical to what has been done before.
>
> Thanks for your comments! We think the construction of cross docking benchmark is very necessary and meaningful, since cross docking can better evaluate the capacity of various docking methods in a more practical scenario compared to self-docking, while there is no corresponding benchmark before PoseX, thus the publish of PoseX could be greatly beneficial to the research of models and algorithms of AI-docking.
>
> >Weakness 3: The way the authors present their benchmarks' results (as well constructed as they are) is ideal for a scientific journal but less common for a machine learning venue. For example, with the font size and scale of Figure 2, it is hard to quickly discern which machine learning method performs "best" in the self and cross-docking contexts (with all caveats in mind). Maybe highlighting the "most notable" results with a box around them would be a way to address this? Also, the methods in Figure 2 are unorganized (i.e., without category labels), so it is hard for a machine learning person (without expertise) to know which method is a model based purely on deep learning and which is a physics-based method.
>
> Many thanks for your valuable suggestions, and we have update Figure 2 and Figure 3 according to your advice.
>
> >Weakness 4: Why did the authors select Astex Diverse as a benchmark dataset instead of the PoseBusters Benchmark dataset (which is arguably more well-known now and less prone to show up in each method's training set)?
>
> Thanks for your comments! Actually PoseX is an extension of PoseBuster, specifically: (1) introducing cross-docking benchmarking to evaluate the capacity of various docking methods on cross docking scenario; (2) including recently published protein-ligand complex crystal structures data for both self-docking (PoseX-SD) and cross-docking (PoseX-CD) (2022.01.0.1 - 2025.01.01); while PoseBuster only focus on self-docking scenario and the data was published from 2021.01.01 to 2023.05.30 (Data leakage exists with the latest AI methods). As for the evaluation on Astex, we just followed the setup of PoseBuster, where both Astex and PosetBuster datasets are all evaluated.
>
> >Question 1: How extensible is PoseX's codebase? For example, how hard is it for current users to build on top of the benchmark to add new datasets or prediction tasks for evaluation? Do you have any case studies in this direction?
>
> The PoseX codebase was designed with strong extensibility in mind. It not only provides the complete construction pipeline for PoseX-SD and PoseX-CD, but also includes a user-friendly script (dataset/create_custom_set.py) that allows users to generate a custom dataset simply by supplying items of {PDB_ID}_{CCD_ID}. Any dataset built with this script can be directly fed into the existing evaluation pipeline without further modification. For example, the Astex Diverse set reported in the paper was itself reconstructed using this script, ensuring it fully conforms to the evaluation protocol.

---

> ### Comment · Reviewer_yfAx · 2025-11-22
> **Response to rebuttal**
>
> I would like to thank the authors for their thoughtful rebuttals. My main question remaining is why the authors (seem) to only report self-docking results for the Astex Diverse dataset from the PoseBusters paper rather than the self-docking PoseBusters Benchmark dataset from the PoseBusters paper. I understand that both of these (self-docking) datasets show up in the training datasets of recent AI-based methods. However, some recent work has shown that the PoseBusters Benchmark dataset contains more diverse (and perhaps more challenging) types of protein-ligand interactions compared to the Astex Diverse dataset.

---

> ### Author Response · Authors · 2025-11-23
>
> Thanks for your discussion! we did not report the self-docking performance on PoseBuster since data in PoseBuster are overlapped with the training data of some recently published methods such as AlphaFold3 and Boltz-1, which makes the evaluation and the comparison less rigorous and unfair to other methods. While there is no overlap between our PoseX-SD and the training data of all the evaluated methods, which makes the evaluation and the comparison more rigorous and fair to all the methods. Actually, PoseX-SD is the new version of PoseBuster where recently published new crystal structures in PDB were included (2022.01.0.1 - 2025.01.01) while the data processing procedure was kept the same to PoseBuster. Therefore, we consider that there is no need to evaluate on PoseBuster since we have already evaluated on PoseX-SD.

---

> > ### Comment · Reviewer_yfAx · 2025-11-23
> > **Response to Authors' Comment**
> >
> > Thank you for clarifying this detail. This evaluation makes more sense to me as a result.

---

> > > ### Author Response · Authors · 2025-11-24
> > >
> > > Thanks again for your great effort in reviewing this paper and the suggestions are so valuable that we would like to polish the benchmark according to your advices in future work. We wonder that whether you could raise the score if all the concerns have been addressed?

---

> > > > ### Comment · Reviewer_yfAx · 2025-11-24
> > > > **Increased score**
> > > >
> > > > I have raised my score accordingly.

---

> > > > > ### Author Response · Authors · 2025-11-25
> > > > >
> > > > > We really appreciate all the effort you have put in, thank you again for your hard work on reviewing our paper!

---

### Official Review · Reviewer_xw3v · 2025-10-23

**Soundness:** 3
**Presentation:** 3
**Contribution:** 3
**Rating:** 6
**Confidence:** 4

**Summary:**

The paper introduces PoseX, an open-source benchmark for evaluating self-docking and cross-docking across 23 methods spanning physics-based docking, AI docking, and AI co-folding. It analyzes pocket-similarity effects and a standardized post-processing relaxation pipeline that mitigates steric clashes. On both self- and cross-docking, several AI methods outperform physics-based tools on top-1 success. Relaxation often improves PB-Valid while not fixing chirality. A public leaderboard is announced.

**Strengths:**

+ The scope is important. The benchmark emphasizes cross-docking, which better reflects real-world use than self-docking alone, and covers 23 diverse methods.
+ The evaluation is informed. Success combines RMSD ($<2 \AA$) with PB-Valid to penalize chemically implausible poses, and analyses of relaxation and chirality are informative.
+ A leaderboard and an automated relaxation pipeline (e.g., OpenMM-based) improve accessibility and reproducibility.

**Weaknesses:**

- It is unclear whether relaxation and other post-processing steps are applied consistently and fairly across method categories, since many
  physics tools already perform internal minimization. Please list which methods received additional OpenMM relaxation and justify comparability.
- The current success definition is top-1 only and couples RMSD $< 2 \AA$ with PB-Valid. Reporting top-$k$ (e.g., $k = 5,10$) pose-ranking metrics, and separate PB-Valid vs. RMSD breakdowns with confidence intervals would better reflect screening use and decouple geometry from plausibility.
- Pocket-similarity is computed against pre-2022 pockets to avoid training overlap, but more detail is needed to ensure ligand- or
  pocket-level leakage is not reintroduced. Please provide formal leakage checks and family-level analyses.

**Questions:**

Please refer to the above weaknesses.

---

> ### Author Response · Authors · 2025-11-21
>
> >Weakness 1: It is unclear whether relaxation and other post-processing steps are applied consistently and fairly across method categories, since many physics tools already perform internal minimization. Please list which methods received additional OpenMM relaxation and justify comparability.
>
> Thanks for your comments! For all AI-based approaches (both AI docking methods and AI co-folding methods) evaluated in our work, only DynamicBind employs relaxation process, and we have already disabled it in our evaluation. As for physics based approach, OpenMM based similar conformation optimization was adopted in Glide, Glide (IFD) and Discovery Studio, while these internal optimization steps are not auxiliary options but are structurally embedded in the algorithms’ search and scoring processes.
>
> >Weakness 2: The current success definition is top-1 only and couples RMSD < 2Å with PB-Valid. Reporting top-k (e.g., k=5, 10) pose-ranking metrics, and separate PB-Valid vs. RMSD breakdowns with confidence intervals would better reflect screening use and decouple geometry from plausibility.
>
> Thanks for your valuable feedback. Firstly, we only reported top-1 performance in our work, which just follow the setup of previous works such as PoseBuster and PoseBench, where also only top-1 results were reported. Furthermore, there are also some methods such as EquiBind and UMD V2 that could not give top-k results. But we have to acknowledge that the top-k evaluation is significant and deserve to be explored, while it is indeed a a huge project where there is a lot of work to do. We are glad to extend our benchmark to support top-k evaluation in the future to fill gaps in this problem to make contribution to the research community. As for the decoupling of RMSD and PB-Valid, we have already included the independent performance of RMSD with the form of mean value ± standard deviation of three independent runs in Table S4, which could be an alternative of confidence interval analysis to some extent, and the independent analysis of PB-Valid in Figure S8 and Figure S9, where specific indicators of PB-Valid were demonstrated, and we also supplemented a column “PB-Valid” in the end of previous Figures for overall summary of PB-Valid failure-rate.
>
> >Weakness 3: Pocket-similarity is computed against pre-2022 pockets to avoid training overlap, but more detail is needed to ensure ligand- or pocket-level leakage is not reintroduced. Please provide formal leakage checks and family-level analyses.
>
> We thank the reviewer for raising this important point. We restricted the data in PoseX to crystal complex structures released between January 1, 2022 and January 1, 2025 (Section 3.3), which avoids overlap with the training data of all evaluated AI methods that listed in Table S3. Therefore, all the protein-ligand complexes in PoseX are completely unseen at the complex/structure level—guaranteeing zero complex-level leakage.
>
> Regarding ligand-level leakage, we agree that many ligands in new crystal structures are chemically identical or highly similar to the ligands that appeared in pre-2022 complexes, which mainly due to the exact ligand sets are difficult to determined precisely since training data processing procedures are different for different models. This makes precise ligand-level leakage quantification challenging or even unavoidable when using real PDB data. However, since docking performance (especially cross-docking) is heavily influenced by the specific protein conformation, using strictly post-cutoff crystal structures remains the most rigorous and practical way to minimize memorization effects while preserving realistic chemical space.
>
> For pocket-level leakage, we explicitly computed pocket structural similarity (TM-score) against all pockets in the entire pre-2022 PDB. In section 4.2.2, we introduced the calculation details of pocket similarity and conducted statistical analysis for different similarity levels (Figure 3, S4, S5).
>
> Regarding family-level analysis, because pocket structure is the primary determinant of protein function and ligand recognition, the pocket similarity distribution in Figure S1b already serves as a proxy for family-level similarity.
>
> We believe these clarifications, combined with the temporal split and the quantitative analysis that provided in Figure S1b and Table S3, fully address the leakage concerns and further strengthen confidence in the validity of our benchmark. Thank you again for your valuable feedback that helps strengthen the paper.

---

> ### Author Response · Authors · 2025-11-27
> **Reminder for Discussion**
>
> Thanks for your valuable comments and suggestions, and we have already responded to your questions. We wonder that whether we have addressed all your concerns? We are looking forward to your response. Thanks a lot!

---

### Official Review · Reviewer_DUgZ · 2025-10-30

**Soundness:** 3
**Presentation:** 2
**Contribution:** 2
**Rating:** 4
**Confidence:** 4

**Summary:**

An open-source benchmark that evaluates self-docking (SD) and cross-docking (CD) across 23 methods (physics-based, AI docking, and AI co-folding), using RMSD and PoseBusters validity (PB-Valid), plus an OpenMM-based relaxation module. The authors report that 9 AI methods surpass GNINA on CD (vs only 3 on SD), and that performance in CD strongly depends on pocket similarity (TM-score of residues within 10 Å, compared to pre-2022 pockets).

PoseX curates 718 SD and 1,312 CD entries (2022–2025 PDB releases), applies filtering (e.g., ligand/protein proximity, symmetry-mate removal), clusters by sequence, aligns conformers by alpha carbons, and in CD evaluates a ligand against other conformers of the same target; success is top-1 RMSD < 2Å (optionally with PB-Valid), averaged at target level to correct for uneven per-target counts.

Compared to the previous benchmarks, it has well-designed data processing workflow, leaderboard, and extensive comparison.

**Strengths:**

- **Timely scope & coverage**. Large-scale comparison over 23 methods spanning physics-based, AI docking, and AI co-folding models; inclusion of commercial tools improves external validity. Also, the authors used dataset with PDBs deposited after 2022.
- **Cross-docking emphasis**. Clearly motivates CD as the practical setting and provides a concrete construction.

**Weaknesses:**

- **Pocket-conditioning fairness**. CD results show pocket-conditioned AI docking models (e.g., SurfDock, UMD-V2, DiffDock-Pocket) against co-folding models that do not receive an explicit pocket mask. However, as pocket information is already known as very important information in predicting binding structure, I think comparing co-folding models' structure prediction performance (Figure 2) without pocket information with AI docking models are not a fair comparison. Furthermore, Pocket-based methods operate inside a pre-specified cavity/box, so even when the true pocket is dissimilar (apo$\to$holo shift, induced fit), their search space, and thus the worst-case ligand RMSD, is tightly bounded by the pocket box. Co-folding models, by contrast, must first localize the site over the entire protein surface/volume and then place the ligand, facing an orders-of-magnitude larger search space and unbounded RMSD penalties for site mislocalization. Consequently, direct RMSD comparisons systematically favor pocket-conditioned docking, and are not information-matched or fair.

- **Missing confidence interval**. The paper reports overall means (and references Table S4/Fig. 2), but it does not present thorough confidence intervals, per-target bootstraps, or seed-sensitivity analyses; important because many methods sample multiple poses and CD averages at the target level. It is therefore hard to tell whether SD$\leftrightarrow$CD deltas are robust to sampling/seed counts and the number of receptor conformers considered.

- **Per-method relaxation settings unclear**. While the OpenMM relaxation is described, it is not fully explicit whether built-in refinements for methods that include their own relaxation were disabled or double-applied, which can tilt PB-Valid and RMSD outcomes. Appendix B lists settings but the policy for overriding internal refinement isn’t crisply stated.

- **Novelty of relaxation module**. The introduction of PB-Valid and OpenMM-based relaxation is a helpful addition, but it is not clear that these components represent a substantial methodological contribution. They appear to extend existing evaluation or refinement tools rather than introduce fundamentally new ideas.

**Questions:**

- **Ground truth in CD is geometry-only**. CD "truth" is defined by transferring the ligand after receptor alignment; there is no check that interaction patterns are conserved. Did authors consider any kinds of interaction pattern comparison such as interaction-fingerprint between co-crystal structure and CD structures?
- **Data-processing thresholds need justification**. How did authors choose the criteria of 0.2A in intermolecular distances in selection step (Table S1,2)?
- **How did authors run DynamicBind?** Although DynamicBind needs apo protein structure, authors did not clarify any information about this.
- **Relaxation policy**. For methods that already implement internal relaxation/refinement, did you disable their internal step before applying your OpenMM relaxation, or apply both?

---

> ### Author Response · Authors · 2025-11-21
>
> >Weakness 1: Pocket-conditioning fairness. CD results show pocket-conditioned AI docking models (e.g., SurfDock, UMD-V2, DiffDock-Pocket) against co-folding models that do not receive an explicit pocket mask. However, as pocket information is already known as very important information in predicting binding structure, I think comparing co-folding models' structure prediction performance (Figure 2) without pocket information with AI docking models are not a fair comparison. Furthermore, Pocket-based methods operate inside a pre-specified cavity/box, so even when the true pocket is dissimilar (apo→holo shift, induced fit), their search space, and thus the worst-case ligand RMSD, is tightly bounded by the pocket box. Co-folding models, by contrast, must first localize the site over the entire protein surface/volume and then place the ligand, facing an orders-of-magnitude larger search space and unbounded RMSD penalties for site mislocalization. Consequently, direct RMSD comparisons systematically favor pocket-conditioned docking, and are not information-matched or fair.
>
> Thanks for pointing out this, and we could not agree more with you. Actually, we have already noticed the gap between the two classes of methods and labeled each method with the pocket information in Table 2 (column “Pocket Required”), and we also have presented the performance analysis on whether the pocket is specified in Figure S10 in Appendix.
>
> >Weakness 2: Missing confidence interval. The paper reports overall means (and references Table S4/Fig. 2), but it does not present thorough confidence intervals, per-target bootstraps, or seed-sensitivity analyses; important because many methods sample multiple poses and CD averages at the target level. It is therefore hard to tell whether SDCD deltas are robust to sampling/seed counts and the number of receptor conformers considered.
>
> We thank the reviewer for the careful suggestion on statistical robustness. Two points may need clarification:
>
> 1. All results are already reported as mean ± standard deviation of three independent runs (explicitly stated in Figure 2 caption and fully listed in Table S4). For nearly all methods, std is below 1% (95% of cases) or 2% (100% of cases), indicating low seed sensitivity.
> 2. Per-target averaging is adopted to explicitly address the uneven number of receptor conformers per target in PoseX-CD : success rate is first computed within each target and then averaged across targets (Section 4.1 ).
>
> >Weakness 3: Per-method relaxation settings unclear. While the OpenMM relaxation is described, it is not fully explicit whether built-in refinements for methods that include their own relaxation were disabled or double-applied, which can tilt PB-Valid and RMSD outcomes. Appendix B lists settings but the policy for overriding internal refinement isn’t crisply stated.
>
> Thanks for your comments! To our knowledge, for all AI-based approaches (both AI docking methods and AI co-folding methods) evaluated in our work, only DynamicBind employs relaxation process, and we have already disabled it in our evaluation, and we have also clarified this point in the manuscript (Appendix B.2.7).
>
> >Weakness 4: Novelty of relaxation module. The introduction of PB-Valid and OpenMM-based relaxation is a helpful addition, but it is not clear that these components represent a substantial methodological contribution. They appear to extend existing evaluation or refinement tools rather than introduce fundamentally new ideas.
>
> First of all, PB-Valid is proposed by PoseBuster [1], it is not proposed by us, we just use it to evaluate the physicochemical validity and structural plausibility of the generated binding poses, as previous works [1][2] always do. As for the relaxation module, compared to the previous version in DynamicBind, we have optimized the process such as: (1) we retain water molecules and metal ions within a 5 Å range of the ligand; (2) we address the lack of bond order information in PDBFixer (please refer to Appendix C for details). We consider that all of these are more likely to be a engineering-level or tool-level contribution, rather than a methodology contribution.
>
> [1] Buttenschoen, Martin, Garrett M. Morris, and Charlotte M. Deane. "PoseBusters: AI-based docking methods fail to generate physically valid poses or generalise to novel sequences." *Chemical Science* 15.9 (2024): 3130-3139.
>
> [2] Ma, Sihan, et al. "PoseBench: Benchmarking the robustness of pose estimation models under corruptions." *arXiv preprint arXiv:2406.14367* (2024).

---

> ### Author Response · Authors · 2025-11-21
>
> >Question 1: Ground truth in CD is geometry-only. CD "truth" is defined by transferring the ligand after receptor alignment; there is no check that interaction patterns are conserved. Did authors consider any kinds of interaction pattern comparison such as interaction-fingerprint between co-crystal structure and CD structures?
>
> Thanks for the valuable suggestion. protein-ligand interaction (PLI) fingerprint comparison is indeed a meaningful complementary metric, which we are now computing using ProLIF, and the corresponding results will be included in the appendix in the revised manuscript.
>
> >Question 2: Data-processing thresholds need justification. How did authors choose the criteria of 0.2A in intermolecular distances in selection step (Table S1,2)?
>
> The construction processes of both PoseX-SD and PoseX-CD, including the step “the criteria of 0.2A in intermolecular distances” you mentioned, are mainly referred to PoseBuster[1], which is a frequently used high-quality dataset used for self-docking evaluation.
>
> [1] Buttenschoen, Martin, Garrett M. Morris, and Charlotte M. Deane. "PoseBusters: AI-based docking methods fail to generate physically valid poses or generalise to novel sequences." *Chemical Science* 15.9 (2024): 3130-3139.
>
> >Question 3: How did authors run DynamicBind? Although DynamicBind needs apo protein structure, authors did not clarify any information about this.
>
> All AI-docking models, including DynamicBind, were run on the holo structures provided by the PoseX dataset, exactly as defined for SD and CD tasks (Figure 1 and Tables S1–S2). This ensures that the input follows the SD and CD setups. We have supplemented the clarification in Appendix B.2.7 in the revised manuscript.
>
> >Question 4: Relaxation policy. For methods that already implement internal relaxation/refinement, did you disable their internal step before applying your OpenMM relaxation, or apply both?
>
> Thanks for your comments! To our knowledge, for all AI-based approaches (both AI docking methods and AI co-folding methods) evaluated in our work, only DynamicBind employs relaxation process, and we have already disabled it in our evaluation, and we have also clarified this point in the manuscript (Appendix B.2.7).

---

> ### Author Response · Authors · 2025-11-27
> **Reminder for Discussion**
>
> Thanks for your valuable comments and suggestions, and we have already responded to your questions. We wonder that whether we have addressed all your concerns? We are looking forward to your response. Thanks a lot!

---

> > ### Comment · Reviewer_DUgZ · 2025-11-27
> >
> > I thank the authors for their detailed response and clarification.
> >
> > First, I apologize for overlooking the statistical details provided in Table S4.
> > I appreciate you pointing this out.
> >
> > However, I still have significant concerns regarding the fairness of the comparison and the lack of results for the cross-docking (CD) interaction analysis.
> >
> > [W1] Fairness of Comparison (Pocket vs. Blind): I appreciate that the authors marked the pocket usage in Table 2. However, I remain convinced that simply labeling this distinction is insufficient given the claims made in the text. For instance, stating that "SurfDock (78.4%) achieves the overall state-of-the-art performance" implies a direct superiority over other methods, including co-folding models that solve a fundamentally harder problem.
> > To ensure a fair evaluation, I strongly suggest that the leaderboard and text be reorganized to: Separate "Pocket-Given" and "Blind/Global" methods into distinct evaluation tracks.
> > Compare co-folding models primarily against other blind methods, or use their pocket-specified versions (if available) when comparing against methods like SurfDock.
> > Comparing these distinct strategies in a single ranking without clearer segmentation creates a misleading narrative about model performance.
> >
> > [Q1] Interaction Fingerprints (ProLIF): I strongly think that interaction pattern preservation is critical for assessing Cross-Docking success (especially for physics-based methods). Could you please provide a summary or preliminary results of the ProLIF analysis?
> >
> > [Q2] While I understand the intent to maintain a uniform input format across the benchmark, DynamicBind is explicitly designed to handle Apo structures and predict the Apo-to-Holo transition. Forcing Holo inputs may constitute an out-of-distribution setting for this model, potentially failing to capture its intended utility or unfairly penalizing its performance. I suggest either testing it with Apo structures (as intended) or explicitly discussing this as a significant limitation of the current benchmarking protocol for dynamic-aware models.
> >
> > I will finalize my decision pending the inclusion of the interaction fingerprint results and a more concrete plan to address the fairness/grouping of the leaderboard.

---

> ### Author Response · Authors · 2025-11-28
>
> > [W1] Fairness of Comparison (Pocket vs. Blind): I appreciate that the authors marked the pocket usage in Table 2. However, I remain convinced that simply labeling this distinction is insufficient given the claims made in the text. For instance, stating that "SurfDock (78.4%) achieves the overall state-of-the-art performance" implies a direct superiority over other methods, including co-folding models that solve a fundamentally harder problem. To ensure a fair evaluation, I strongly suggest that the leaderboard and text be reorganized to: Separate "Pocket-Given" and "Blind/Global" methods into distinct evaluation tracks. Compare co-folding models primarily against other blind methods, or use their pocket-specified versions (if available) when comparing against methods like SurfDock. Comparing these distinct strategies in a single ranking without clearer segmentation creates a misleading narrative about model performance.
>
> Thanks for your valuable feedback on the comparison fairness between pocket-given and blind docking methods, and we could not agree more with you. Follow your suggestions, we have reorganized the presentation in **Section 4.2.1** ("Overall Performance Analysis") by supplementing the performance analysis for the “Pocket-Given” track (docking with specified pocket) and the “Blind-Docking” track (docking without specified pocket) respectively, which is analyzed with blue text and illustrated with Figure 3. We believe that the revision could address your concerns on the fairness of comparison.
>
>
> > [Q1] Interaction Fingerprints (ProLIF): I strongly think that interaction pattern preservation is critical for assessing Cross-Docking success (especially for physics-based methods). Could you please provide a summary or preliminary results of the ProLIF analysis?
>
> We evaluate PLIF-Valid [1] for all methods and provide results in the appendix (**Table S5 and Figure S10**). PLIF-Valid assesses recovery rates of protein-ligand interaction fingerprints (computed using ProLIF). Overall, the ranking of methods on PLIF-Valid closely follows the docking success rate ranking, with SurfDock achieving the best performance. The relaxation module consistently improves PLIF-Valid for nearly all methods. Notably, physics-based methods show better interaction preservation than their geometric success rates suggest (e.g., PoseX-SD: DiffDock-L success rate 47.1% > Glide(IFD) 46.5%, but PLIF-Valid 51.7% < Glide(IFD) 55.2%; PoseX-CD: DiffDock success rate 47.1% > Glide IFD 44.8%, but PLIF-Valid 49.9% < Glide IFD 58.8%), likely due to their implicit consideration of protein–ligand interactions.
>
> [1] Errington, D., Schneider, C., Bouysset, C., & Dreyer, F. A. (2025). Assessing interaction recovery of predicted protein-ligand poses: D. Errington et al. *Journal of Cheminformatics*, *17*(1), 76.
>
> > [Q2] While I understand the intent to maintain a uniform input format across the benchmark, DynamicBind is explicitly designed to handle Apo structures and predict the Apo-to-Holo transition. Forcing Holo inputs may constitute an out-of-distribution setting for this model, potentially failing to capture its intended utility or unfairly penalizing its performance. I suggest either testing it with Apo structures (as intended) or explicitly discussing this as a significant limitation of the current benchmarking protocol for dynamic-aware models.
>
> Thanks for your comments! Just as what you said, there is an approximation for DynamicBind evaluation in our work where holo structures were employed as input to maintain a uniform input format across the benchmark rather than the apo structures which were supposed to be. To be rigorous according to your advice, we have discussed the limitation of current benchmarking protocols and setups for DynamicBind (in **Section 6**, "Limitation and Future Work"), which is highlighted with blue color, and we will supplement the corresponding apo structure based evaluation in future work.

---

### Official Review · Reviewer_Az7C · 2025-10-31

**Soundness:** 3
**Presentation:** 2
**Contribution:** 3
**Rating:** 6
**Confidence:** 4

**Summary:**

PoseX is a new benchmark for a new variant in docking task using a series of new "co-folding" methods. Compared to previous benchmarks, PoseX have more design related to the recent changes in docking methods.

**Strengths:**

Since the development of AlphaFold3 and related methods, this task has drawn wildly attention. Old benchmark like PoseBusters show limitation in such tasks. Compared to traditional docking benchmark, cross-docking is harder, and able to detect more hallucinations cases which is often existed in AlphaFold 3 like models.

**Weaknesses:**

There is little to show the difference between co-folding methods and traditonal methods, since co-folding methods might have strong advantages in docking on protein with native conformations. I suggest authors to add more examples and highlight these tasks including docking on AlphaFold3 predicted native conformation or solved native conformations to stress this difference.

**Questions:**

1. Authors can report more details about other affecting factors, including the docking performance affected by molecule size, hard-ness of protein target, etc.
2. The detailed parameters of the molecular dynamics simulation should be showed, which force field, the time length, the maximum round and the strategy, treatment of hydrogen bond, restrictions, solvent information and solvent model, etc.

---

> ### Author Response · Authors · 2025-11-21
>
> >Weakness 1: There is little to show the difference between co-folding methods and traditonal methods, since co-folding methods might have strong advantages in docking on protein with native conformations. I suggest authors to add more examples and highlight these tasks including docking on AlphaFold3 predicted native conformation or solved native conformations to stress this difference.
>
> Thank you for your comments! According to your suggestion, we have already supplemented evaluation on docking with AlphaFold3 predicted conformations, and we could see the results next week. Do you have any other suggestions else on this problem?
>
> >Question 1: Authors can report more details about other affecting factors, including the docking performance affected by molecule size, hard-ness of protein target, etc.
>
> Thanks a lot and it’s a good suggestion! We have supplemented the docking performance affected by molecular weight in Figure S6 (PoseX-SD) and Figure S7 (PoseX-CD), where most methods exhibit a obvious drop (in RMSD) for ligands heavier than 450 Da. As for the influence of protein target hardness, please refer to the analysis on pocket similarity in Figure S4 (PoseX-SD) and Figure S5 (PoseX-CD).
>
> >Question 2: The detailed parameters of the molecular dynamics simulation should be showed, which force field, the time length, the maximum round and the strategy, treatment of hydrogen bond, restrictions, solvent information and solvent model, etc.
>
> The specific parameters for the relaxation method are described in Appendix C.

---

> > ### Author Response · Authors · 2025-11-27
> > **Reminder for Discussion**
> >
> > Thanks for your valuable comments and suggestions, and we have already responded to your questions. We wonder that whether we have addressed all your concerns? We are looking forward to your response. Thanks a lot!

---

> ### Author Response · Authors · 2025-12-01
>
> We have supplemented the performance analysis in Table 1, where the results of docking with AlphaFold3 predicted conformations are compared to the results of the original version in our work, as what you suggested. Here we select two representative AI docking methods for primary analysis, and we could supplement the results of more methods in the camera-ready version of the manuscript if it is necessary.
>
> **Table 1:** Docking performance on PoseX-CD (crystal receptors) versus PoseX-CD* (AlphaFold3-predicted conformations).
> |Methods|RMSD < 2Å (w/o relax.)|RMSD < 2Å (w/ relax.)|RMSD < 2Å & PB-Valid (w/o relax.)|RMSD < 2Å & PB-Valid (w/ relax.)|
> |:---|:----:|:----:|:----:|:----:|
> |DiffDock|45.42±1.01|47.15±1.13|18.83±0.88|46.07±1.27|
> |DiffDock*|43.90±0.68|45.62±0.62|18.88±0.91|44.92±0.51|
> |FABind|24.72±0.20|29.38±0.41|4.00±0.02|21.15±0.58|
> |FABind*|26.23±0.42|30.98±0.37|4.57±0.24|23.48±0.48|
>
> Methods marked with "*" use AlphaFold3-predicted protein structures as receptor input.

---

### Author Response · Authors · 2025-12-01
**Summary to Area Chair**

**Summary of the Revision**

We would like to thank the AC and reviewers for your time and efforts on reviewing our paper, and the valuable comments and suggestions have helped us significantly improve the quality of this work in several directions. Specifically, the main changes in the revised manuscript include:

- We supplemented the docking performance affected by molecular weight in Figure S6 (PoseX-SD) and Figure S7 (PoseX-CD) as suggested by reviewer *Az7C*;
- We supplemented  the performance analysis for the “Pocket-Given” track (docking with specified pocket) and the “Blind-Docking” track (docking without specified pocket) respectively, as suggested by reviewer *DUgZ*;
- We evaluated PLIF-Valid for all methods as the supplementary of RMSD and PB-Valid (Table S5 and Figure S10), as suggested by reviewer *DUgZ*;
- We enhanced the visualization by supplementing a column “PB-Valid” to summarize the overall PB-Valid failure-rate, as suggested by reviewer *xw3v*;
- We beautified Figure 2 and Figure 3 with category labels for better visualization, as suggested by reviewer *yfAx*;
---

**Summary of Our Contributions**

We curated **a new dataset** with newly released protein-ligand complex crystal structures focusing on both **self-docking** and **cross-docking**, and incorporated **23 docking methods** across three main research lines (*physics-based methods*, *AI docking methods*, and *AI co-folding methods*) to make an **exhaustive evaluation**.

For **self-docking**, we constructed PoseX-SD, which is an *extension of PoseBuster* possessing **more data** (*308 entries for PoseBuster / 718 entries for PoseX-SD*) and **newer release date** of crystal structures (*2021.01.01 - 2023.05.30 for PoseBuster / 2022.01.0.1 - 2025.01.01 for PoseX-SD*).

Compared to self-docking, **cross-docking** is more practical in drug discovery. However, there is currently no comprehensive dataset for the evaluation of various methods. In our study, we constructed PoseX-CD (*date: 2022.01.0.1 - 2025.01.01 / 1312 entries*), which is the **first** benchmarking and **fills a critical gap in this area**.

---

**Summary of Reviewers' Replies**

Reviewer *yfAx* expressed appreciation for our revisions and have accordingly **raised their scores to 6**;

Reviewer *DUgZ* gave very positive feedback on our first-round response and the final score is expected to be improved after supplementing the analysis results that the reviewer want to see (the original wording: “**I will finalize my decision pending the inclusion of the interaction fingerprint results and a more concrete plan to address the fairness/grouping of the leaderboard.**”).

Reviewer *Az7C* and reviewer *xw3v* did not response while we have carefully addressed all their concerns and updated the manuscript;

We deeply regret about the information leakage incident which totally throws off our rhythm since we really spent a significant amount of work on rebuttal and the subsequent ensuing discussion. We kindly request that AC could take all our discussions into account as all these were happened before the information leakage incident.

Finally, we thank all the reviewers again for their insightful comments, and we sincerely thank the AC for the time and efforts to review our work. We believe that the revised manuscript is much more improved and the work could be greatly beneficial to the research of models and algorithms of AI-docking and help the evaluation be more practical.

---

### Meta-Review · Area_Chair_fYap · 2026-01-04

**Summary:**

The paper introduces PoseX, a comprehensive benchmark for protein-ligand docking that focuses on both self-docking and, crucially, cross-docking. The authors curated a new dataset comprising 718 self-docking and 1,312 cross-docking entries derived from PDB structures released between 2022 and 2025 to avoid training data leakage. The benchmark evaluates 23 methods across three categories: physics-based, AI docking, and AI co-folding. Key contributions include an automated relaxation module to resolve steric clashes and a public leaderboard. The study concludes that AI methods generally outperform physics-based methods in docking success rates, though AI co-folding methods struggle with ligand chirality.

**Reviewer Concerns:**

## Addressed:

- Reviewer noted that comparing pocket-conditioned AI models directly against blind co-folding models was unfair. The authors separated to distinct Pocket-given and Blind-docking tracks in the revised manuscript and leaderboard.
- Reviewer requested interaction fingerprint analysis to ensure physical plausibility beyond simple RMSD. The authors integrated PLIF-Valid  into the appendix, as the supplementary of RMSD.
- Reviewer questioned the novelty compared to PoseBench and why the Astex dataset was used. The authors clarified that PoseX focuses on cross-docking and uses a strict post-2022 temporal split to ensure zero training leakage.
- Reviewer requested comparisons using AlphaFold3-predicted structures. The authors added this one.

## Outstanding:
- Reviewer DUgZ noted that DynamicBind is designed for Apo structures, but the benchmark uses Holo structures (OOD). The authors acknowledged this limitation and added a discussion in the paper, promising Apo evaluation in future work. While not fully resolved experimentally, the acknowledgment is sufficient.

**Reviewer Scores:**

Reviewer Az7C: 6 ==> 6. Conerns are addressed and keep positive.


Reviewer DUgZ: 4 ==> 6 Concerns are addressed and likely move to positive.


Reviewer xw3v: 6 ==> 6  Concerns are addressed.


Reviewer yfAx: 4 -> 6 Concerns are addressed and likely move to positive.

---

### Decision · Program_Chairs · 2026-01-26

Accept (Poster)